# Health Outcomes, Pathogenesis and Epidemiology of Severe Acute Malnutrition (HOPE-SAM): rationale and methods of a longitudinal observational study

Mutsa Bwakura-Dangarembizi,[1] Beatrice Amadi,[2] Claire D Bourke,[3] Ruairi C Robertson,[3] Benjamin Mwapenya,[4] Kanta Chandwe,[2] Chanda Kapoma,[2] Kapula Chifunda,[2] Florence Majo,[4] Deophine Ngosa,[2] Pamela Chakara,[4] Nivea Chulu,[2] Faithfull Masimba,[4] Idah Mapurisa,[4] Ellen Besa,[2] Kuda Mutasa,[4] Simutanyi Mwakamui,[2] Thompson Runodamoto,[4] Jean H Humphrey,[4] Robert Ntozini,[4] Jonathan C K Wells,[5] Amee R Manges,[6] Jonathan R Swann,[7] A Sarah Walker,[8] Kusum J Nathoo,[1] Paul Kelly,[2,3] Andrew J Prendergast,[3,4] the HOPE-SAM study team

For numbered affiliations see end of article.

**Correspondence to**
Dr Mutsa Bwakura-Dangarembizi;
mbwakura@medsch.uz.ac.zw

## ABSTRACT

**Introduction** Mortality among children hospitalised for complicated severe acute malnutrition (SAM) remains high despite the implementation of WHO guidelines, particularly in settings of high HIV prevalence. Children continue to be at high risk of morbidity, mortality and relapse after discharge from hospital although long-term outcomes are not well documented. Better understanding the pathogenesis of SAM and the factors associated with poor outcomes may inform new therapeutic interventions.

**Methods and analysis** The Health Outcomes, Pathogenesis and Epidemiology of Severe Acute Malnutrition (HOPE-SAM) study is a longitudinal observational cohort that aims to evaluate the short-term and long-term clinical outcomes of HIV-positive and HIV-negative children with complicated SAM, and to identify the risk factors at admission and discharge from hospital that independently predict poor outcomes. Children aged 0–59 months hospitalised for SAM are being enrolled at three tertiary hospitals in Harare, Zimbabwe and Lusaka, Zambia. Longitudinal mortality, morbidity and nutritional data are being collected at admission, discharge and for 48 weeks post discharge. Nested laboratory substudies are exploring the role of enteropathy, gut microbiota, metabolomics and cellular immune function in the pathogenesis of SAM using stool, urine and blood collected from participants and from well-nourished controls.

**Ethics and dissemination** The study is approved by the local and international institutional review boards in the participating countries (the Joint Research Ethics Committee of the University of Zimbabwe, Medical Research Council of Zimbabwe and University of Zambia Biomedical Research Ethics Committee) and the study sponsor (Queen Mary University of London). Caregivers provide written informed consent for each participant. Findings will be disseminated through peer-reviewed journals, conference presentations and to caregivers at face-to-face meetings.

### Strengths and limitations of this study

► Rigorous collection of longitudinal data on morbidity, mortality and nutritional status during inpatient care and for 48 weeks after initial admission for severe acute malnutrition (SAM) in HIV-positive and HIV-negative children.
► Laboratory substudies investigating enteropathy, microbiota, metabolomics and immune cell function provide a unique opportunity to understand which pathogenic pathways contribute to SAM and whether these processes normalise with nutritional rehabilitation.
► High loss to follow-up due to participants returning to home settings following hospital discharge.
► The clinical heterogeneity of the study participants, including comorbidities such as infections, may make it challenging to identify the specific causes of clinical outcomes.
► Potential bias in recruiting well-nourished controls only from hospitals will be reduced by inclusion of community-based controls, including well-nourished siblings of children with SAM.

## INTRODUCTION

Malnutrition underlies almost half of all childhood deaths in developing countries.[1] Severe acute malnutrition (SAM) is defined by a weight-for-height Z-score (WHZ) <–3, mid-upper arm circumference (MUAC) <115 mm and/or bilateral pitting

oedema.[2] Current treatment guidelines distinguish two groups: (i) children with uncomplicated SAM who can be managed in the community and (ii) children with complicated SAM, who are hospitalised and undergo resuscitation, stabilisation and nutritional rehabilitation. In-hospital mortality in children with complicated SAM remains high despite the implementation of WHO guidelines.[3] Furthermore, SAM presents as two major clinical phenotypes: non-oedematous SAM (marasmus), characterised by severe wasting, and oedematous SAM (kwashiorkor), a more complex syndrome characterised by bilateral pitting oedema, steatosis and diarrhoea.[4 5] Despite differing clinical outcomes, treatment protocols are the same for both oedematous and non-oedematous SAM.

A contributory factor to high inpatient mortality is the co-occurrence of HIV infection in around one-third of children hospitalised for SAM in sub-Saharan Africa.[6 7] While new HIV infections in children have declined,[8] a substantial number of infected children are diagnosed late and present with malnutrition. There is also a growing population of HIV-exposed uninfected (HEU) children who have immune abnormalities, poor growth and higher risk of mortality and infectious morbidity.[9] Hence, HIV has transformed the epidemiology and outcomes of SAM.[10] Even with standardised treatment approaches, inpatient deaths are almost fourfold higher among HIV-positive children with SAM (herein termed HIV-SAM) compared with HIV-negative children with SAM (30.4% vs 8.4%), for reasons that remain unclear[10]; this mortality is threefold higher than would be expected from anthropometric parameters alone.[10] Management of HIV-SAM is particularly challenging because HIV fundamentally alters the clinical presentation of malnutrition and the response to treatment. Children with HIV-SAM are more stunted and wasted; have a higher frequency of persistent diarrhoea; tend to have delayed nutritional recovery and have a more complicated clinical course than HIV-negative children with SAM.[10]

## Long-term outcomes of SAM
Following resolution of complications and return of appetite, children are discharged from hospital to continue therapeutic feeds at home. However, emerging data indicate high post discharge mortality following in-hospital management of SAM.[11–13] Malnutrition, together with young age, HIV infection and pneumonia, has been associated with higher post discharge mortality.[14] One of the largest prospective studies of growth and mortality in children with SAM (FuSAM), conducted in Malawi from July 2006 to March 2007, collected 12-month outcome data on 87% of 1024 children admitted to the nutrition ward.[11] A total of 427 (42%) died and 44% of these deaths occurred after discharge from hospital. Survival was greatest among those who were nutritionally cured on discharge from outpatient therapeutic feeding centres, defined as two consecutive visits with >80% expected weight-for-height, no oedema and clinically stable. The risk of mortality after hospital discharge was fourfold higher for HIV-SAM compared with HIV-negative children with SAM, but the outcomes among HEU children were not reported. The loss to follow-up was high in the FuSAM study because there was only one follow-up visit, 1 year after discharge from outpatient-feeding centres. A recent study from Kenya identified malnutrition and HIV infection as key drivers for post discharge mortality, with 52% of deaths attributable to MUAC <11.5 cm and 11% to HIV infection.[15]

The impact of SAM appears to persist beyond the first year after discharge from hospital. The ChroSAM study, which followed children with SAM 7 years post discharge, showed that children had poorer growth, body composition and physical function compared with siblings and community controls, which are all indicators of future cardiovascular and metabolic disease.[12]

While anthropometry is used to assess nutritional recovery after discharge from hospital, the pattern and quality of growth recovery following SAM is poorly understood. The observation that children treated for SAM have a deficit in lean tissue despite regaining weight suggests that assessing body composition in addition to anthropometry may help to identify children who have not completely recovered and are at potential risk of long-term metabolic diseases.[12] Children with HIV-SAM appear to have potential for catch-up growth in weight-for-age and/or weight-for-height, which have been shown to normalise with treatment even prior to widespread availability of Antiretroviral Therapy (ART)[16]; by contrast, height-for-age shows less potential for catch-up growth.[17] However, the body composition of children with HIV-SAM compared with HIV-negative children with SAM has not been described. Whether children recover fat mass at the expense of lean mass is unknown, but differences in tissue accretion patterns may have implications for survival and long-term metabolic health.[18 19] There is also a need to consider the effect of SAM on the size of body parts which grow at different rates: relatively shorter legs, for example, are associated with epidemiological risk of overweight, coronary artery disease, liver dysfunction and diabetes.[20 21]

Taken together, there is clearly an elevated risk of mortality among HIV-positive children with SAM compared with HIV-negative children with SAM, and an ongoing mortality risk among all children with SAM that persists after discharge from hospital. There are several gaps in our understanding of the long-term outcomes: (i) causes of death have not been clearly defined, (ii) no studies have systematically and longitudinally collected morbidity and mortality data or documented repeat hospitalisations post discharge and (iii) the long-term outcomes of HIV-positive children with SAM in the era of ART availability are unclear.

## Pathogenesis of SAM
Better understanding the pathogenesis of SAM may help to explain the high mortality of children both during and

after hospitalisation and identify new targets for interventions to supplement existing treatment strategies. Consistent evidence that immune mediators are altered in malnutrition[22] and that systemic and intestinal inflammation are associated with poor outcomes in SAM[23] suggests that immune dysfunction contributes to infectious susceptibility.[24] Malnutrition is also characterised by a complex derangement in gut microbial,[25] metabolic,[26] immune[27] and hormonal pathways, organ dysfunction and micronutrient deficiencies in the context of co-infections, enteropathy and chronic inflammation. Several studies have recently provided insights into these perturbations using new tools,[25 26 28 29] including metabolomics and metagenomics, but we still lack a clear understanding of many of the pathogenic pathways driving malnutrition, the interactions between these pathways and which are the most tractable targets for intervention.

SAM shares several pathological and clinical features with HIV, which may explain clinical outcomes in these co-occurring conditions: (i) both are characterised by intestinal damage, leading to impairment of the mucosal barrier and increased intestinal permeability; (ii) both have underlying systemic immune activation; and (iii) both are frequently complicated by persistent diarrhoea, pneumonia and sepsis that may plausibly arise due to loss of intestinal barrier function.[30] Understanding the overlapping impact of HIV and SAM is critical to inform additional interventions to improve outcomes of children with HIV-SAM.

## OBJECTIVES OF HOPE-SAM

The Health Outcomes, Pathogenesis and Epidemiology of Severe Acute Malnutrition (HOPE-SAM) study has two primary objectives:

1. To describe the short-term and long-term clinical outcomes of children with complicated SAM, with and without HIV infection, and to identify the risk factors at admission and discharge from hospital that independently predict these outcomes.
2. To better characterise the pathogenesis of SAM through nested laboratory substudies evaluating

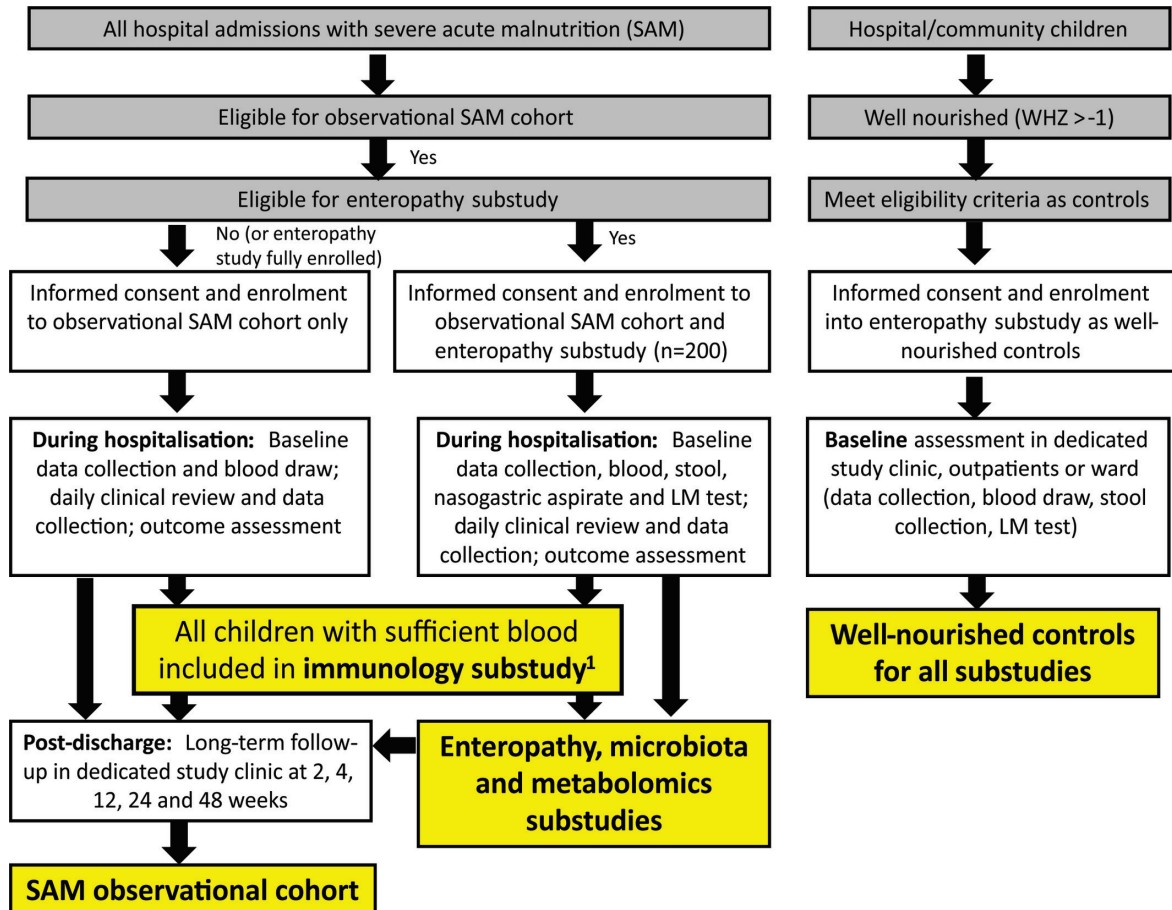

**Figure 1** Study flow chart. All hospital admissions are screened for eligibility for the observational cohort and enteropathy substudy, with procedures undertaken as shown in the flow chart during hospitalisation and post discharge. Well-nourished children from outpatient clinics and the community meeting eligibility criteria as well-nourished controls are enrolled and undergo a single baseline assessment as shown. The immunology, microbiota and metabolomics substudies enrol children as shown. All children with severe acute malnutrition (SAM), regardless of which arm of the study they are enrolled into, are followed for 48 weeks post discharge. [1]The immunology substudy started from 1 June 2017 and required children to have a blood sample >2 mL to conduct cellular assays. LM, lactulose-mannitol; WHZ, weight-for-height Z-score.

enteropathy, gut microbiota, metabolomics and immune cell function.

## STUDY DESIGN

HOPE-SAM is a longitudinal observational cohort study, enrolling between 600–800 children aged 0–59 months admitted with complicated SAM to the tertiary paediatric wards at two sites in Zimbabwe (Parirenyatwa Hospital and Harare Children's Hospital) and one in Zambia (University Teaching Hospital, Lusaka). Both HIV-positive and HIV-negative children will be enrolled. Throughout this paper, 'SAM' refers to all children, regardless of HIV status; where analyses specifically compare children by HIV status, groups are identified as HIV-positive children with SAM (HIV-SAM) and HIV-negative children with SAM. All participants with SAM are followed for 48 weeks post discharge, with longitudinal data collection and blood sampling. The study contains four nested substudies as shown in figure 1. A subgroup of children will be recruited to the enteropathy substudy for which they will have the same follow-up procedures but more intensive biological specimen collection including stool (all time points), urine after lactulose-mannitol (LM) challenge as an assessment of intestinal permeability and nasogastric aspirate (baseline only); these children are also included in microbiota and metabolomics substudies. Children with SAM for whom blood samples are available are included in the immunology substudy, for which circulating inflammatory mediators will be assayed; functional cellular immunology assays will be conducted for all children in the immunology substudy with sufficient sample volume (>2 mL) recruited after June 2017. A group of healthy children recruited from the same hospitals and communities, who are well-nourished and matched to children in the enteropathy substudy by age and HIV status, will have data and specimens collected to provide normative data for the laboratory substudies; these well-nourished controls will not be followed longitudinally.

The study protocol, data collection forms and standard operating procedures are available online at https://osf.io/29uaw/.

## RECRUITMENT

### Screening

Caregivers of all hospitalised children are sensitised about the study. All new admissions aged 0–59 months are screened for SAM, which is defined according to WHO criteria as any of WHZ <–3, MUAC <115 mm (if aged 6–59 months) and/or bilateral pitting oedema. All children with SAM are recruited from hospital, and this study therefore focuses on complicated SAM; children with uncomplicated SAM will not be enrolled.

### Eligibility for observational cohort

All children with SAM whose caregivers are willing to provide written informed consent and to learn their child's HIV status are offered enrolment. Any children who die prior to study enrolment and those with a known malignancy are ineligible.

### Eligibility for enteropathy substudy

Children with SAM aged 6–59 months with a nasogastric tube in place (or due to be placed) are categorised into four groups (HIV-positive oedematous (group A -I); HIV-positive non-oedematous (group A-II); HIV-negative oedematous (group C-I) and HIV-negative non-oedematous (group C-II)) as shown in table 1. Children meeting eligibility criteria will be enrolled throughout the study recruitment period until sufficient specimens have been collected from the groups shown in table 1. Children in the enteropathy substudy are stratified into age bands (6–11 months, 12–23 months and 24–59 months) to enable age-matching of well-nourished controls. Children with underlying chronic gastrointestinal disease or a known malignancy are ineligible.

### Eligibility for microbiota and metabolomics substudies

Children enrolled into the enteropathy substudy are also included in the microbiota and metabolomics substudies since these substudies use the stool, urine and plasma samples collected for enteropathy analyses.

### Eligibility for immunology substudy

The immunology substudy comprises all children with SAM (drawn from both the observational cohort

| Table 1 Enteropathy substudy groups | | | |
|---|---|---|---|
| | SAM* | | Well-nourished controls WHZ>–1 |
| Children aged 6–59 months | Oedematous† | Non-oedematous | |
| HIV-positive (HIV PCR+ if <18 months; HIV antibody+ if >18 months) | n=50 (group A-I) | n=50 (group A-II) | n=100 (group B) |
| HIV-negative (HIV PCR– if<18 months; HIV antibody – if>18 months) | n=50 (group C-I) | n=50 (group C-II) | n=100 (group D) |

Note that children below 6 months of age are excluded from the enteropathy substudy to avoid interrupting exclusive breast feeding during the lactulose-mannitol test.

*SAM defined according to WHO criteria.

†Presence of bilateral pitting pedal oedema.

SAM, severe acute malnutrition; WHZ, weight-for-height Z-score.

and the enteropathy substudy, as shown in figure 1) providing a blood sample of sufficient volume (>2 mL) for cellular assays after 1 June 2017.

## Well-nourished controls

Controls are children drawn from the same hospitals and communities as cases with SAM (including well-nourished sibling controls), who are aged 6–59 months (matched to enteropathy substudy children within age bands), well-nourished (WHZ >−1) and clinically well (no acute illness or current infections) with known HIV status. Controls are categorised into two groups: well-nourished HIV-positive (group B) and well-nourished HIV-negative (group D), as shown in table 1. Children with underlying chronic gastrointestinal disease or a known malignancy are ineligible. Well-nourished controls provide comparison biomarker data for all the laboratory substudies.

## STUDY PROCEDURES

Study procedures are outlined in table 2.

| Table 2 | Summary of procedures in observational cohort | | | | | | |
|---|---|---|---|---|---|---|---|
| | **Hospitalisation** | | **Post discharge*** | | | | |
| **Assessment** | **Baseline†** | **Discharge‡** | **2 weeks** | **4 weeks** | **12 weeks** | **24 weeks** | **48 weeks** |
| Caregiver informed consent to join observational cohort | x | | | | | | |
| Summary checklist | x | | | | | | |
| Locator information§ | x | | | | | | |
| Acute admission information | x | | | | | | |
| Baseline data | x | | | | | | |
| Daily clinical review¶ | Daily during hospitalisation | | | | | | |
| Blood collection** | x | x | | | x | x | x |
| HIV testing†† | x | | | | | | |
| CD4 count and viral load (HIV-infected children only) | x | | | | x | x | x |
| Full blood count‡‡ | x | x | | | x | x | x |
| Anthropometry | x | x | x | x | x | x | x |
| Skinfold thickness§§ | | x | x | x | x | x | x |
| Body composition¶¶ | x | x | x | x | x | x | x |
| Discharge data collection | | x | | | | | |
| Daily morbidity diary | Daily during follow-up period by caregivers | | | | | | |
| Follow-up clinic: history, examination, morbidity and mortality data | | | x | x | x | x | x |

*Windows will be created around these post-discharge time-points to maximize follow-up for caregivers who miss visits or are unavailable, as follows: 2 weeks (optimal window from 1 week to 2 weeks + 6 days; allowable window up to 2 weeks + 6 days); 4 weeks (optimal window 3-5 weeks, allowable window up to 9 weeks + 6 days); 12 weeks (optimal window 10-14 weeks, allowable window up to 19 weeks + 6 days); 24 weeks (optimal window 20-28 weeks, allowable window up to 43 weeks + 6 days); 48 weeks (optimal window 44-52 weeks, allowable window up to 71 weeks + 6 days). †Children will be enrolled as soon as possible after hospitalisation and will undergo baseline investigations as soon as possible after enrolment. This is to provide a window of opportunity to time collection of research specimens with clinical specimens, and to ensure that the child is clinically stable before undertaking research investigations.
‡The discharge procedures will be undertaken on the day of discharge or as close as possible to that date.
§Locator information will be updated at subsequent visits if caregivers have moved or changed contact details.
¶A clinical review will be undertaken every day between admission and discharge by the study clinician.
**5.4 mL of blood (depending on child weight; amount will not exceed 2 mL/kg total over 2-week period) will be collected by a study nurse into endotoxin-free EDTA tubes. Samples will be used to store whole blood, Peripheral Blood Mononuclear Cells (PBMC) and plasma for subsequent measurement of C reactive protein and albumin. Where blood sample volumes allow (≥2 mL sample), bacterial binding assays and whole blood stimulations will be conducted and culture supernatants and cells stored for subsequent assessment of immune cell function at each time point. Study blood samples will not be collected from children with known haemoglobin <60g/L.
††HIV testing is conducted as part of routine clinical practice, but if it has not been undertaken, the study sample will be used to test for HIV, as stated in the informed consent form, since HIV status is required to allocate children to study groups.
‡‡Full blood count results will be transcribed from clinical records; if not done by clinical teams, the EDTA sample will be used to measure Full Blood Count (FBC) in clinical laboratories at each site.
§§Skinfold thickness (triceps, subscapular, supra-iliac) and mid-thigh circumference will be measured using Holtain callipers or tape measure.
¶¶Body composition will be assessed by bioimpedance vector analysis.

## Baseline procedures

Baseline data on maternal and household characteristics, the child's medical history and current illness are collected by a study nurse. Anthropometry, including body composition measured by whole-body (wrist-ankle) bio-electrical impedance analysis (BodyStat 1500MD; BodyStat, UK), leg length using an electronic knemometer (Zimbabwe only, due to availability of knemometers) and triceps, subscapular and supra-iliac skinfold thickness using callipers (Holtain, Crymych, UK), are undertaken at baseline. Blood (1 mL/kg up to 5.4 mL maximum) is collected at baseline into an endotoxin-free EDTA tube for all children and, in the enteropathy substudy, additionally into a PAXgene tube (PreAnalytiX, Hombrechtikon, Switzerland) for subsequent transcriptomic analysis. Blood is not collected from children with severe anaemia (known haemoglobin <60 g/L). HIV testing is carried out in accordance with national guidelines as part of routine clinical practice; where it has not been done, the child's HIV status is ascertained using a rapid test antibody algorithm for children over 18 months, or HIV DNA PCR for children under 18 months. CD4 count/percentage and viral load are measured in HIV-positive children. Maternal HIV status is documented where available, so that HIV-exposed uninfected children can be identified. Blood samples are sent to research laboratories at each site to conduct whole blood stimulation and bacterial binding assays (as described in the immunology substudy) and to store aliquots of whole blood, peripheral blood cells and plasma at –80ºC.[31] In the enteropathy substudy, nasogastric aspirate, stool and urine (after an oral dose of lactulose and mannitol) are also collected. Lactulose and mannitol are ingested by the child after fasting and urine is collected over a 2-hour period to measure recovery of lactulose and mannitol, a measure of intestinal absorptive capacity and permeability, as previously described.[32]

## Daily procedures

Routine inpatient management is undertaken by ward clinical teams according to local hospital protocols, which are based on WHO guidelines.[2 33] In addition, the HOPE-SAM study clinician at each hospital site collects daily data until discharge on clinical parameters (including daily examination), resolution of acute infections, nutritional recovery (loss of oedema, restoration of appetite, weight gain) and treatment/nutritional supplements received; this will allow us to evaluate differences in management between countries. Children with HIV-SAM who are ART-naive start ART according to national guidelines, which are based on WHO recommendations.[2 34]

## Discharge

The clinical team decides when the child is ready to be discharged, which is generally when their medical complications are resolving and the child has a good appetite and is clinically well and alert.[2] Children receive ready-to-use therapeutic feeds (RUTF) to take at home according to local guidelines. At discharge, the study nurse collects data and a repeat blood sample (including full blood count) and undertakes discharge anthropometry, body composition, leg length (Zimbabwe only) and triceps, subscapular and supra-iliac skinfold thickness measurements (table 2). The caregiver is given a daily morbidity diary and pre-prepared stickers corresponding to different illnesses and shown how to complete the diary. The caregiver is provided with the date of the first follow-up appointment and contact details of the study nurse.

## Follow-up

Children attend follow-up appointments at dedicated study clinics at 2, 4, 12, 24 and 48 weeks post discharge. At each visit, the study physician undertakes a clinical assessment and the study nurse captures illness, medication and feeding data. Clinic data are transcribed from handheld medical records if available and the morbidity diary is reviewed and a new diary and stickers supplied. Anthropometry, body composition, leg length (Zimbabwe only) and triceps, subscapular and supra-iliac skinfold thicknesses are measured at each visit. Acute illnesses are treated in the study clinic or the child is referred to hospital if necessary. Children with relapsed malnutrition are provided with nutritional supplements or RUTF according to local guidelines or readmitted to hospital if they develop complicated SAM. Transport reimbursement for clinic attendance is provided to caregivers for each visit.

Blood is collected at weeks 12, 24 and 48 post discharge into endotoxin-free EDTA tubes to measure full blood count, CD4 count and viral load (HIV-positive children only), conduct whole blood stimulation and bacterial binding assays (where blood volumes > 2 mL) and store peripheral blood cells and plasma aliquots for subsequent analyses (all blood samples), including soluble and cellular markers of immune activation, as outlined in online supplementary table 1. Children in the enteropathy substudy have additional stool and urine collection following lactulose-mannitol dosing as shown in table 3.

Caregivers are reminded of follow-up visits by phone, and visit completion is tracked on a dedicated database. If caregivers do not attend follow-up appointments, attempts are made to contact them by phone and home visits are made if feasible, particularly for those defaulting on the 48-week visit, so that long-term outcome data can be collected. For post discharge deaths, a home visit is undertaken by study nurses where possible to conduct a verbal autopsy. Children who are readmitted to one of the study sites with relapsed SAM have data collected during the new episode of hospitalisation. The study ends for each participant at the week 48 visit.

**Table 3** Summary of procedures for cases in the enteropathy substudy

| Assessment | Hospitalisation | | Post discharge* | | | | |
| --- | --- | --- | --- | --- | --- | --- | --- |
| | Baseline† | Discharge‡ | 2 weeks | 4 weeks | 12 weeks | 24 weeks | 48 weeks |
| Caregiver informed consent to join observational cohort and enteropathy substudy | X | | | | | | |
| Summary checklist | X | | | | | | |
| Locator information§ | X | | | | | | |
| Acute admission information | X | | | | | | |
| Baseline data | X | | | | | | |
| Daily clinical review¶ | Daily during hospitalisation | | | | | | |
| Blood collection** | X | X | | | X | X | X |
| HIV testing†† | X | | | | | | |
| CD4 count and viral load (HIV-infected children only) | X | | | | X | X | X |
| Full blood count‡‡ | X | X | | | X | X | X |
| Gastric aspirate§§ | X | | | | | | |
| Stool collection¶¶ | X | X | | | X | X | X |
| Lactulose-mannitol testing*** | X | X | | | X | | X |
| Anthropometry | X | X | X | X | X | X | X |
| Skinfold thickness††† | | X | X | X | X | X | X |
| Body composition‡‡‡ | X | X | X | X | X | X | X |
| Discharge data collection | | X | | | | | |
| Daily morbidity diary | Daily during follow-up period by caregivers | | | | | | |
| Follow-up clinic: history, examination, morbidity and mortality data | | | X | | X | X | X |

*Windows will be created around these post-discharge time points to maximise follow-up for caregivers who miss visits or are unavailable, as follows: 2 weeks (1–3 weeks); 4 weeks (3–5 weeks); 12 weeks (10–14 weeks); 24 weeks (20–28 weeks) and 48 weeks (44–52 weeks).
†Children will be enrolled within 24 hours of hospitalisation and will undergo baseline investigations within 72 hours of hospitalisation. This is to provide a window of opportunity to time collection of research specimens with clinical specimens, and to ensure that the child is clinically stable before undertaking research investigations.
‡The discharge procedures will be undertaken on the day of discharge or as close as possible to that date.
§Locator information will updated at subsequent visits if caregivers have moved or changed contact details.
¶Daily clinical review will be conducted every day between admission and discharge by the study clinician.
**During hospitalisation, 5.4 mL of blood (depending on child weight; amount will not exceed 2 mL/kg total over 2-week period) will be collected by a study nurse into a 2.7 mL endotoxin-free EDTA tube and a 2.7 mL PAXGene tube, for subsequent isolation of RNA and gene expression analysis. After discharge (weeks 12, 24 and 48), 5.4 mL of blood (depending on child weight; amount will not exceed 2 mL/kg total over 2-week period) will be collected by a study nurse into two 2.7 mL endotoxin-free EDTA tubes.
††HIV testing is conducted as part of routine clinical practice, but if it has not been undertaken, the study sample will be used to test for HIV (see section 9.4), as stated in the informed consent form, since HIV status is required to allocate children to study groups.
‡‡Full blood count results will be transcribed from clinical records; if not done by clinical teams, the EDTA sample will be used to measure FBC in clinical laboratories at each site.
§§A gastric juice sample will be collected at the same time as the blood draw by aspirating the nasogastric tube with a sterile feeding syringe, to test for gastric pH; sterile water or saline will then be instilled and a sample of gastric juice collected for storage for subsequent PCR and culture (section 7.5.2).
¶¶Stool collection will be undertaken at the same time as the blood draw.
***Lactulose-mannitol (LM) testing will be conducted, with collection of a baseline urine sample, followed by a 2-hour urine collection post-LM ingestion. This test will be deferred until children are judged to be clinically stable by the study physician during daily reviews. In general, this will be a child in the nutritional rehabilitation phase, who has no cardiorespiratory compromise.
†††Skinfold thickness (triceps, subscapular, supra-iliac) and mid-thigh circumference will be measured using Holtain callipers or tape measure.
‡‡‡Body composition will be assessed by bioimpedance vector analysis.

## SUBSTUDIES

As outlined in figure 1, four nested substudies will use biological specimens to address mechanistic questions related to enteropathy, microbiota, metabolomics and immune function.

## Enteropathy substudy

The gut, which acts as an internal interface between humans and the environment, must contain the nutrient stream and the symbiotic microbiota while allowing molecular intimacy to permit absorption. The mechanism underlying this duality is the integrity of the gastrointestinal barrier; intestinal damage (enteropathy) can impair this critical barrier function. A spectrum of enteropathies affect children in developing countries.[30] Environmental enteric dysfunction (EED), characterised by small intestinal inflammation, blunted villi and increased intestinal permeability, is almost universal and is morphologically indistinguishable from HIV enteropathy.[30] Children in

resource-poor settings also suffer from frequent diarrhoea, food insecurity and micronutrient deficiencies, which all exacerbate enteropathy.[30] As a result, a cycle of intestinal infection, impaired mucosal function and malnutrition commonly arises, which may ultimately precipitate SAM, especially in the context of HIV infection.[35 36] It is not yet established if the enteropathy seen in children with SAM,[37] which we here refer to as malnutrition enteropathy,[37] is qualitatively or quantitatively distinguishable from EED. In addition to local intestinal pathology, enteropathies may cause systemic pathology due to persistent immune activation arising from enteric inflammation and microbial translocation across the damaged gut wall.[30] It is becoming apparent that chronic inflammation may be particularly deleterious in malnourished individuals[23]; in children with SAM, systemic inflammation arising from underlying enteropathy may further increase morbidity and mortality.

We hypothesise that (i) the degree of enteropathy during hospitalisation differs between oedematous and non-oedematous SAM and is independently associated with morbidity, mortality and nutritional recovery during hospitalisation; (ii) the degree of enteropathy at discharge is independently associated with morbidity, mortality and relapse of SAM; and (iii) children with HIV-SAM have more severe enteropathy than HIV-negative children with SAM, which contributes to their poorer outcomes.

Using stored samples, a longitudinal series of investigations will compare gastric and small intestinal barrier function, using a range of biomarkers to capture the domains of malnutrition enteropathy (online supplementary table 2). To understand better the extra-intestinal consequences of enteropathy, we will first compare the microbial composition of the upper gut and plasma using deep sequencing in a subgroup of children with paired gastric and blood samples. Second, we will undertake transcriptomics using PAXGene blood samples to determine (i) whether there are differences in gene expression profiles between well-nourished controls, HIV-negative children with SAM and HIV-positive children with SAM (including comparison of oedematous and non-oedematous types); and (ii) whether specific patterns of gene expression are associated with morbidity and mortality in SAM.

### Microbiota substudy

Normal assembly of the gut microbiota in early life is critical for many aspects of physiological, neurological and immune development.[38] Recent evidence suggests that an immature or pathogenic microbiota plays a causative role in the pathogenesis of SAM.[25] For example, a number of microbial taxa have been identified, including *Faecalibacterium prausnitzii*, which discriminate and predict gut microbiota maturity and child growth.[28] Other pathogenic microorganisms, including IgA-targeted *Enterobacteriaceae*, are associated with impaired growth and may contribute to SAM.[39] Nutritional rehabilitation with RUTF induces temporary recovery of a disturbed microbiota; however, the microbiota appears to revert back to an immature

diseased state following nutritional recovery.[29] HIV infection is also associated with a disturbed gut microbiota,[40] which may further compound enteropathy phenotypes. Furthermore, there is some evidence that differences exist in malnutrition enteropathy between oedematous and non-oedematous SAM[41]; however, few studies have investigated differences in the gut microbiota between the two forms of the disease.

We hypothesise that (i) a unique gut microbial signature exists in HIV-SAM, compared with HIV-negative children with SAM, that is independently associated with morbidity, mortality, nutritional recovery and degree of enteropathy during hospitalisation; (ii) a unique gut microbial signature exists in oedematous compared with non-oedematous SAM; (iii) specific microorganisms or gut microbial diversity indices are independently associated with morbidity, mortality, nutritional recovery and degree of enteropathy during hospitalisation; and (iv) the gut microbiota is partially restored to a healthy state with nutritional rehabilitation but reverts to a dysbiotic state during follow-up, which predicts morbidity, mortality and relapse of SAM.

Using stored stool samples collected at baseline, a cross-sectional investigation will determine differences in the gut microbial composition and predicted function between: HIV-negative children with SAM versus HIV-SAM, oedematous versus non-oedematous SAM and well-nourished controls. Gut microbial composition and predicted function will be compared between groups at discharge and at 12, 24 and 48 weeks post discharge. Briefly, total DNA and/or RNA will be extracted from stool samples and used as template for next-generation sequencing library preparation and for quantitative PCR. Whole metagenome shotgun sequencing will be performed using the HiSeq 2500 system. Raw metagenomic sequencing data will be quality-filtered and analysed through a well-validated bioinformatics pipeline using MetaPhlAn[42] and HUMAnN.[43] The compositional and predicted functional metagenomic data generated will be used to identify signatures of SAM and to investigate associative links between specific gut microbial signatures and clinical outcomes.

### Metabolomics substudy

During SAM, metabolic processes are altered in response to a starved environment and may plausibly contribute to long-term clinical outcomes. Previous studies suggest that amino acid turnover, lipid metabolism, oxidative stress and other metabolic pathways are disrupted in SAM and may be associated with disease state and clinical outcome[26 44 45]; however, little is known about how the metabolic phenotype responds to nutritional therapy. It is hypothesised that disturbed gut microbiota composition and function may drive microbial metabolic dysregulation in addition to host-derived dysregulation. Of particular interest are differences in the metabolic phenotype between oedematous and non-oedematous SAM. The 'reductive adaptation' seen in non-oedematous

SAM (utilisation of fat and muscle stores) is disrupted in oedematous SAM, which may contribute to differences in clinical outcomes. Specifically, protein turnover, inflammation, oxidative stress and bile acid metabolism are disrupted in oedematous-SAM, which may contribute to comorbidities including diarrhoea, steatosis and enteropathy.[46 47]

We hypothesise that (i) a unique plasma and urine metabolic phenotype exists in oedematous compared with non-oedematous SAM during hospitalisation, which is independently associated with morbidity, mortality and nutritional recovery; (ii) the metabolic phenotype is partially restored to a healthy state with nutritional rehabilitation but reverts to a disturbed state during follow-up, which predicts morbidity, mortality and relapse; and (iii) both host-derived and gut microbial-driven metabolic dysregulation underlie clinical outcomes.

Using stored urine and plasma samples collected during hospitalisation, a cross-sectional investigation will determine differences in the metabolic phenotype between children with oedematous SAM, non-oedematous SAM and well-nourished controls. Urine and plasma metabolic phenotypes will be compared between groups at discharge and 12, 24 and 48 weeks post discharge. Briefly, global untargeted metabolomic phenotyping will be performed via $^1$H nuclear magnetic resonance (NMR) spectroscopy using a 700 MHz Bruker NMR spectrometer to identify metabolic signatures of SAM. Targeted analysis via ultra-performance liquid chromatography-mass spectrometry will be performed to examine specific pathways of interest, including tryptophan and bile acid metabolism.

## Immunology substudy

Bacterial infections are common among children hospitalised for SAM[23 48–50] and mortality is driven by a range of species,[48 50–53] consistent with generalised defects in innate antibacterial defence. Increased infectious morbidity and mortality persist after discharge from hospital,[10 17 54] suggesting that restoration of antibacterial immune responses may lag behind nutritional rehabilitation. A recent randomised trial in children with SAM confirmed that deaths following hospitalisation were predominantly due to bacterial infections but were not prevented by daily co-trimoxazole prophylaxis.[17] Collectively, these observations highlight that children remain vulnerable to infection despite current treatment approaches; targeting persistent immune dysfunction could plausibly reduce infectious mortality after discharge.[24]

Multiple innate and adaptive immune mediators are dysregulated in malnutrition.[24 27 55] However, few studies have assessed cellular immune function in malnourished children; most existing studies were undertaken decades ago on small cross-sectional cohorts without the benefit of recent advances in immunology techniques.[27] Immune dysfunction in SAM likely reflects both *intrinsic* defects,

whereby immune cells lack capacity to adequately respond to infection, and *extrinsic* defects, where cells have intact antibacterial capacity but are chronically modulated by the systemic pro-inflammatory environment which characterises SAM (ie, heightened pro-inflammatory cytokines[44] and circulating bacterial antigens[23 56 57]). Systemic inflammation is directly associated with mortality in SAM[23] and driven by multiple comorbidities, including bacterial translocation from the damaged gut into the blood, subclinical infections and metabolic dysregulation.[44 58 59] The implications of innate immune cell dysfunction for subsequent acquisition of infections and infectious mortality have not been investigated.

We hypothesise that (i) antibacterial functions of innate immune cells are compromised in SAM due to a combination of intrinsic and extrinsic defects; (ii) innate immune cell function is independently associated with infectious morbidity and mortality during hospitalisation for SAM; and (iii) nutritional rehabilitation only partly restores innate immune cell function, leading to an ongoing risk of bacterial infections post discharge.

Using blood samples collected at baseline, discharge and 12, 24 and 48 weeks post discharge, the longitudinal relationship between circulating innate immune cell function and bacterial infections will be assessed. The intrinsic phagocytic capacity, secreted cytokine response and maturation state of innate immune cells after culture with bacterial antigens will be assessed. Plasma concentrations of endotoxin and pro-inflammatory mediators will be quantified at each time point and the degree to which these extrinsic factors influence innate immune cell antibacterial function will be assessed via plasma co-culture with innate immune cells from healthy donors. Bacterial infections during hospitalisation will be diagnosed using clinical criteria and blood culture, stool culture and urinalysis where available.

## SAMPLE SIZES
### Observational study

The observational cohort will recruit as many children with SAM as possible during the period of enrolment (July 2016 to March 2018), estimated at 600–800 children (capped at 800 maximum), to assess clinical and nutritional outcomes among HIV-positive and HIV-negative children hospitalised with SAM. Assuming mortality of 15%, overall loss to follow-up of 15% and recruitment target of 800 children, there would be 560 evaluable children at 48 weeks, of whom 224 would have HIV-SAM based on an estimated inpatient HIV prevalence of 40%. This will provide >80% power to detect absolute differences of 17% in binary outcomes between HIV-SAM and HIV-negative children with SAM, and of 0.33 times the SD in continuous outcomes.

### Enteropathy substudy

The sample size was estimated using previously reported values for LM ratios, which remain a widely used

non-invasive marker of enteropathy. Comparing 100 versus 100 children with two-sided alpha=0.025 (to allow for two primary comparisons, ie, HIV-SAM vs HIV-negative children with SAM, and HIV-SAM vs well-nourished HIV-positive children) provides >80% power to detect differences in mean LM ratio during hospitalisation of at least 0.16 (assuming SD 0.36), a difference which would be clinically relevant given the LM ratios previously reported for well-nourished children (0.42), malnourished children (1.3) and children with persistent diarrhoea (2.85) in the Gambia.[60] It also provides >80% power to detect differences of at least 0.1 in the mean change in LM ratio from enrolment (assuming SD for change=0.23 and 7% missing samples). For inflammatory markers, comparing 100 versus 100 children with two-sided alpha=0.025 provides >80% power to detect differences in mean $\log_{10}$ concentrations of at least 0.44 times their SD, or 2.75-fold differences between groups. Inclusion of well-nourished controls provides an indication of normal ranges in young African children. HIV-positive and HIV-negative SAM groups will be stratified to include approximately 50 children with and without oedematous malnutrition, if possible.

### Microbiota and metabolomics substudy

Power calculations are difficult in metagenomics and metabolomic analyses due to the large number of observed outcomes and unknown effect sizes and variance. Previous studies using smaller sample sizes have identified significant taxonomic differences in twin pairs discordant for oedematous-SAM (n=13)[25] and metabolic differences between the two forms of SAM (n=40).[26] These studies suggest that a difference of 50% in metabolites could be expected. Using analysis of covariance, setting α=0.05 and assuming either low (ρ=0.1) or high (ρ=0.7) correlation, the study would require 95–126 subjects to achieve 80% power.[61] False discovery rate multiple correction testing will be applied to reduce the high-dimensionality of the data and limit false positives.

### Immunology substudy

Up to 200 children with SAM and 200 well-nourished controls will be included in a cross-sectional analysis of innate immune cell function during hospitalisation. Assuming similar infectious mortality to a recent Kenyan study (15%),[17] a cohort of 200 provides 80% power to detect associations between immune profiles and infectious mortality at an OR of 1.7 and two-sided alpha of 0.05. We will aim for 100 children with longitudinal analysis of innate immune cell function at discharge, 12, 24 and 48 weeks post discharge.[17]

### STUDY OUTCOMES AND RISK FACTORS

The main study outcomes are clinical (mortality, morbidity and relapse of malnutrition) and nutritional (weight, height, MUAC, leg length, head circumference, mid-thigh circumference, skin-fold

thickness and body composition by bioimpedance vector analysis) assessed over 48 weeks of follow-up. Mortality is assessed in hospital by daily physician review and, post discharge, through study visits and by telephone where possible for children who are lost to follow-up. Morbidity during hospitalisation is assessed through daily clinical assessments and available hospital laboratory tests. Morbidity after discharge is assessed, first, using daily morbidity diaries, in which caregivers record episodes of illness (lethargy interfering with feeding; respiratory distress; diarrhoea; oedema and fever); second, from caregiver recall and review of handheld medical records at each follow-up visit; and, third, from data collected during hospitalisation for children who are readmitted during the follow-up period. Time-to-recovery from malnutrition will be evaluated during hospitalisation; relapse of malnutrition during follow-up will be categorised as moderate acute malnutrition, uncomplicated SAM and complicated SAM, according to WHO definitions. Nutritional outcomes will be expressed both as continuous variables (attained Z-score and change in Z-score between visits), and as categorical variables (moderate wasting, WHZ<−2; severe wasting, WHZ<−3; stunting, HAZ<−2; severe stunting, HAZ<−3; underweight, WAZ <−2; and microcephaly, head circumference-for-age <−2).

Risk factors will be evaluated at baseline, hospital discharge and over the period of follow-up for associations with clinical and nutritional outcomes. In addition to baseline clinical and demographic factors, the following laboratory parameters will be evaluated: haemoglobin, serum albumin, C reactive protein (CRP), CD4 count and HIV viral load (for HIV-positive children). Haemoglobin, CD4 and HIV-viral load will be measured in real time and the results reviewed during follow-up clinics.

Data on potential confounders are collected at baseline, discharge and during the follow-up period, including child feeding practices, household socioeconomic status (defined by household income and cooking method), maternal employment and education, and household factors such as water, sanitation and hygiene practices, availability of electricity, location (rural, peri-urban or urban) and household size.

### ANALYSIS

All analyses will be interpreted exploratively since HOPE-SAM is an observational study with multiple risk factors, outcomes and substudies. For all analyses, p values will not be artificially adjusted, but interpreted as exploring the strength of evidence supporting any association. The only exception is the use of approaches to minimise false discovery when analysing high-dimensional data from the microbiota and metabolomics substudies, as described.

## Observational cohort

The primary comparison will be the clinical and nutritional outcomes of children with HIV-SAM compared with HIV-negative children with SAM. We will review all deaths and adjudicate clinical diagnoses and causes of death to ensure robust and consistent data across sites. We will compare each participant's clinical management to WHO guidelines to identify any contributory factors in hospital care. Factors associated with outcomes during hospitalisation (eg, mortality, nutritional recovery) will be determined for each group (HIV-SAM and HIV-negative children with SAM) using multivariable analysis (Cox models for time-to-event data, linear models for continuous outcomes). Factors associated with outcomes over 48 weeks post discharge (hospital readmission, morbidity and mortality, relapse, anthropometry, body composition and response to ART) will be determined for each group (HIV-SAM and HIV-negative children with SAM) using multivariable analysis (Cox models for time-to-event data, linear models for continuous outcomes). HIV-positive children with SAM and HIV-negative children with SAM will be included in one model together with the risk factors, and interaction tests will be used to investigate whether associations between risk factors and outcomes differ between the two groups of children. We will evaluate the ability of MUAC at discharge to predict long-term outcomes using receiver-operator-characteristic (ROC) analysis, in the whole cohort and within the subgroups of HIV-SAM and HIV-negative children with SAM. We will then evaluate whether addition of other variables improves the predictive capacity of MUAC (using WHO criteria in those >6 months, and published data for children <6 months[62]) for each group, including body composition, haemoglobin, albumin and CRP, plus CD4%, viral load and timing of ART initiation (HIV-SAM only). We will construct multivariable models and compare them with MUAC alone using the net-reclassification index.

## Body composition analysis

Previous work in body composition by bio-electrical impedance in Ethiopian infants and children with SAM has shown that the conventional approach, predicting total bodyweight from height-adjusted impedance, fails due to confounding by oedema.[63] The same project validated an alternative approach, known as Bio-electrical Impedance Vector Analysis, and described significant differences between each of three groups: healthy controls, oedematous-SAM and non-oedematous SAM. Vector analysis splits impedance into two height-adjusted components, resistance and reactance, which are further linked through phase angle (PA). Variability in these components is associated with biochemical parameters.[64] These variables will be explored using graphical analysis or transformed into age-adjusted and sex-adjusted Z-scores for statistical comparison, including longitudinal analyses. Higher PA indicates better nutritional status, while declining height-adjusted resistance over time indicates loss of oedema.

## Enteropathy substudy

The primary comparison for the enteropathy substudy will be between HIV-positive children with SAM (group A) and HIV-negative children with SAM (group C), stratified by presence or absence of oedema. Control groups (B and D) are well-nourished children with or without HIV, to provide normative data for biomarkers and to evaluate the impact of SAM within each HIV group. Thus, biomarkers among HIV-positive children with SAM will first be compared with HIV-negative children with SAM (to evaluate the impact of HIV) and, second, with well-nourished HIV-positive children (to evaluate the impact of SAM). Biomarkers among HIV-negative children with SAM will be compared with well-nourished HIV-negative children. For each continuous outcome, simple descriptive analysis will be used to compare groups during hospitalisation using t-tests on appropriately transformed data. For any outcome with moderate (p<0.05) evidence of difference between either group a regression model will be constructed including groups A, B, C and D to directly test (using interactions) whether there is a synergistic effect of HIV-SAM versus HIV-negative SAM versus HIV alone versus neither. These models will also be used to explore whether there is any evidence for heterogeneity in effects between oedematous and non-oedematous SAM. Associations between enrolment factors (eg, intestinal permeability and microbial translocation) will be explored using pairwise Spearman correlations and principal components analysis. Mean changes at the follow-up time points in each group will be estimated, and groups compared (as above) using generalised estimating equations. For outcomes that differ across SAM groups over time, multilevel models will be used to explore possible predictors from the other factors measured. Time to nutritional recovery will be compared using Kaplan-Meier and log-rank tests, and Cox models to adjust for baseline differences between groups.

## Microbiota and metabolomics substudy

The primary comparison will be between HIV-negative children with oedematous and non-oedematous SAM, with a separate comparison between HIV-positive children with SAM and HIV-negative children with SAM. Analyses will examine (i) differences in metagenomic/metabolomic variables between groups at each time point; (ii) differences in metagenomic/metabolomic variables within groups over time; (iii) correlations between metagenomic and metabolomic variables and (iv) correlations between metagenomic/metabolomic variables and clinical outcomes. A systematic analysis will be undertaken to reduce high-dimensional data, integrate the multi-omics data sets and minimise false discovery.

Compositional metagenomic data will be compared between groups for indices of alpha and beta diversity. Principal coordinate analysis and partial least-squares discriminant analysis will be performed on metabolomics data to identify overall differences between groups. High-dimensional data sets will be reduced using random

forest models to identify taxa, microbial gene families and metabolites that most strongly contribute to differences between groups, corrected by Benjamani-Hochburg false discovery rate detection. Targeted analysis by qRT-PCR will validate differential abundance or expression of candidate microbial genes. Longitudinal comparisons will be performed within and between groups using multilevel simultaneous component analysis. Orthogonal projections to latent structures models will integrate metabolomic and metagenomic data while linear regression, canonical correlation and hierarchal clustering analysis will measure correlations between -omics data sets. Finally, ROC analysis will identify the ability of different analytes to predict long-term nutritional and clinical outcomes.

### Immunology substudy

Integrated profiles of innate immune cell function will be generated for each child using principal components analysis followed by hierarchical clustering.[65 66] This data -reduction method identifies whether absolute levels of specific markers or relative differences between markers differentiate children into groups. The resulting innate immune profiles will be compared between HIV-SAM, HIV-negative children with SAM and well-nourished groups using univariable tests and multivariable analysis of variance of the principal components.

To address the relationship between immune function and infections, regression analyses will determine whether baseline innate immune profiles (or the individual parameters defining them) are associated with the infectious morbidity or mortality during hospitalisation, using logistic models for binary outcomes and linear models for duration. Key clinical characteristics, including age, sex, oedema and baseline WHZ, will be added to models to investigate their confounding effects. Multivariable stacked regression methods will be used to compare the impact of different factors on severe bacterial infections based on heterogeneity tests.

To determine whether treatment for SAM restores innate immune cell antibacterial function, mixed effects regression models will compare longitudinal changes in individual immune parameters, and the principal components calculated from the weights identified at baseline (which include well-nourished controls). Similarities and differences in longitudinal immune profiles will be compared between groups using non-metric multidimensional scaling.[65 67 68] This approach will group children according to their composite innate immune function, allowing the duration and variability of immune restoration to be evaluated over the course of nutritional rehabilitation. Binary logistic regression will determine whether innate immune profiles at discharge are associated with morbidity or mortality during follow-up.

### PATIENT AND PUBLIC INVOLVEMENT

Patients and their caregivers were not involved in the design of the study. During recruitment, all caregivers of children admitted to hospital were given information about the study; those whose children had severe acute malnutrition were approached to give written informed consent. A meeting to disseminate results of the study to participants and their caregivers will be held at the end of the study. An interactive game to engage caregivers in the science underlying malnutrition is being developed in collaboration with experts from the Centre of the Cell, a unique science education centre based at Queen Mary University of London (https://www.centreofthecell.org/).

### SAFETY REPORTING

For all adverse events, the study team will assess expectedness and relatedness to study activities. Since this is an observational study without interventions, we anticipate that the risk is minimal; however, serious adverse events will be reported to local ethical review boards (Medical Research Council of Zimbabwe, and University of Zambia Biomedical Research Ethics Committee) and the study sponsor (Queen Mary University of London) according to their respective guidelines.

### DATA COLLECTION AND MONITORING

Clinical and demographic data are recorded on paper case report forms. All data are checked for completeness and plausibility before data entry and problems flagged for resolution by the clinical team. All data are double-entered onto a dedicated password-protected online study database and any discrepancies resolved. Study participants are identified on electronic databases only by study numbers (assigned at enrolment); no personal identifiers are entered.

### ETHICS AND DISSEMINATION

The study complies with the principles of the Declaration of Helsinki (2008) and is conducted in compliance with the principles of Good Clinical Practice and local regulatory requirements in each country. Ethical approval was obtained from the University of Zambia Biomedical Research Ethics Committee, the Joint Research Ethics Committee of the University of Zimbabwe and the Medical Research Council of Zimbabwe. The ethical review board of the Sponsor, Queen Mary University of London, provided an advisory review of the study. Since this is an observational study, there is no Data and Safety Monitoring Board.

Results will be disseminated through conference abstracts and peer-reviewed publications and discussed with relevant policymakers and programmers. Study findings will be disseminated to families of participants at face-to-face meetings.

## TIME FRAME AND STUDY STATUS

"Enrolment into the SAM cohorts began in July 2016 and ended in March 2018. Enrolment of well-nourished controls is expected to end in March 2019. All participants with SAM will be followed for 48 weeks, with an expected study completion date of March 2019."

## DISCUSSION

HOPE-SAM aims to document the short- and long-term clinical and nutritional outcomes of HIV-positive and HIV-negative children with SAM, and to identify the factors at presentation and at discharge from hospital that independently predict these outcomes. Mechanistic substudies aim to evaluate the contribution of enteropathy, microbiota, metabolome and innate immune cell function to these clinical outcomes. The prevalence of malnutrition in HIV-positive children is as high as 40% in some settings and the challenges of managing this population are well recognised.[69] The WHO protocol on management of SAM aims to reduce case fatality below 10%, but rates as high as 35% are still reported among HIV-positive children.[5 70] No studies have systematically and longitudinally collected morbidity data in HIV-SAM, or documented repeat hospitalisations and mortality after discharge from hospital, particularly in the current era where ART is available on diagnosis. HOPE-SAM will provide a unique opportunity to enrol and follow a cohort of children managed for SAM in three large hospitals across two sub-Saharan African countries at several time points over a 1-year period. Nested longitudinal laboratory substudies aim to better characterise the pathogenesis of SAM in HIV-positive and HIV-negative children, to determine whether pathogenic processes are normalised during nutritional rehabilitation and follow-up, and to identify potential mechanistic pathways. Our ultimate goal is to use the findings generated in this study to inform new intervention approaches that can be evaluated in clinical trials to improve outcomes among children with SAM.

#### Author affiliations
[1]Department of Paediatrics and Child Health, University of Zimbabwe College of Health Sciences, Harare, Zimbabwe
[2]Tropical Gastroenterology and Nutrition Group, University of Zambia, Lusaka, Zambia
[3]Blizard Institute, Queen Mary University of London, London, UK
[4]Zvitambo Institute for Maternal and Child Health Research, Harare, Zimbabwe
[5]UCL Great Ormond Street Institute of Child Health, London, UK
[6]University of British Columbia, Vancouver, British Columbia, Canada
[7]Imperial College London, London, UK
[8]MRC Clinical Trials Unit at UCL, London, UK

**Acknowledgements** The authors thank Phillipa Rambanepasi, Karen Gwanzura and Agatha Muyenga for financial management of the study; Zinah Sorefan and Daniela Azurunwa for study coordination at Queen Mary University of London; Professor Kim Michaelsen for the use of the electronic knemometers and advice on their use; Dr Gemma Buxton for her help in drafting the case report forms used in the study; and the authors remember the hard work of Edith Mukusho who sadly passed away during the HOPE-SAM study. The authors thank the staff and management and members of the Department of Paediatrics and Child Health at the three hospital sites who have made this study possible. They are indebted to the caregivers, families and children who are participating in the HOPE-SAM study.

**Collaborators** Members of the HOPE-SAM study team not listed in the author list: Harare, Zimbabwe: Virginia Sauramba, Adlight Dandadzi, Chipo Kureva, Johnson Mushonga, Eddington Mpofu, Washington Dune, Tafadzwa Chidhanguro, Sibongile Nkiwane, Sandra Rukobo, Margaret Govha, Patience Mashayanembwa, Leah Chidamba, Bernard Chasekwa, Joice Tome, Rachel Makasi, Wellington Murenjekekwa, Theodore Chidawanyika, Blessing Tsenesa, Stephen Moyo, Penias Nyamwino, Pururudzai Simango, Shepherd Seremwe, Lovemore Chingaoma and Sarudzai Kasaru. Lusaka, Zambia: Andreck Tembo, Mary Mpundu, Evelyn Nyendwa, Gwendolyn Nayame, Dalitso Tembo, Sophreen Mwaba, Esther Chilala, Lucy Macwani, Tenzeni Dumba, Miyoba Chipunza, Lydia Kazhila, Temwaninge Gondwe, Dennis Phiri, Mpala Mwanza, Kanekwa Zyambo.

**Contributors** Designed study: MB-D, BA, CDB, RCR, BM, KC, CK, KCh, DN, PC, NC, FM, JW, ARM, JS, ASW, KJN, PK and AJP. Sought funding: MB-D, BA, CDB, RCR, JHH, ARM, JS, ASW, KJN, PK and AJP. Undertaking study: BM, KC, CK, KCh, FM, DN, PC, NC, FM, IM, EB, KM, SM and TR. Study oversight: MB-D, BA, JHH, KJN, PK and AJP. Analysis: MB-D, BA, CDB, RCR, RN, JW, ARM, JS, ASW, KJN, PK and AJP. Wrote first draft of manuscript: MB-D, CDB, RCR and AJP. Critically revised manuscript: all.

**Funding** This work was supported by the Medical Research Council UK (MR/K012711/1), the Wellcome Trust (107634/Z/15/Z to MB-D; 206225/Z/17/Z to CDB, an award funded in partnership with the Royal Society; 206455/Z/17/Z to RCR; and 108065/Z/15/Z to AJP), and a Bio-Resource Grant from the Centre for Genomic Health within the Life Sciences Initiative at Queen Mary University of London.

**Competing interests** None declared.

**Patient consent for publication** Not required.

**Ethics approval** Medical Research Council of Zimbabwe, University of Zambia Biomedical Research Ethics Committee, Queen Mary University of London.

**Provenance and peer review** Not commissioned; externally peer reviewed.

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
