## [Reviewer comments · BMJ Open]

ARTICLE DETAILS

TITLE (PROVISIONAL)	Health Outcomes, Pathogenesis and Epidemiology of Severe Acute Malnutrition (HOPE-SAM): rationale and methods of a longitudinal observational study
AUTHORS	Bwakura-Dangarembizi, Mutsa; Amadi, Beatrice; Bourke, Claire; Robertson, Ruairi; Mwapenya, Benjamin; Chandwe, Kanta; Kapoma, Chanda; Chifunda, Kapula; Majo, Florence; Ngosa, Deophine; Chakara, Pamela; Chulu, Nivea; Masimba, Faithfull; Mapurisa, Idah; Besa, Ellen; Mutasa, Kuda; Mwakamui, Simutanyi; Runodamoto, Thompson; Humphrey, Jean; Ntozini, Robert; Wells, Jonathan C. K.; Manges, Ameer; Swann, Jon; Walker, Sarah; Nathoo, Kusum; Kelly, Paul; Prendergast, Andrew

VERSION 1 – REVIEW

REVIEWER	André BRIEND University of Tampere Medical School, Tampere, Finland, International Health
REVIEW RETURNED	03-Apr-2018

GENERAL COMMENTS	Children with severe acute malnutrition (SAM) have a high mortality even when treated with currently recommended WHO protocol. The reasons for this are not clear, and in depth studies examining possible mechanisms are most welcome. Following the development of ready-to-use therapeutic foods (RUTF) research on SAM management have focused over the last 15 years on how to simplify treatment and make it more accessible to large number of children. This was a priority, but improving treatment of complicated cases is also needed and this requires a better understanding of current treatment limitations. This proposal addresses precisely this important issue. It will provide invaluable information on how to improve survival of these high risk children. This protocol is a result of an impressive team work involving many high level specialists in different areas and at this stage, there is not much to comment about this ongoing study. Two points, however, may be highlighted. 1) Interpreting the results. The whole protocol is based on the comparison of children with SAM, HIV-infected or HIV uninfected with and without oedema, with controls taken among children receiving inpatient or outpatient care at the study site. The rationale for this is clear, patients but also controls are more easily recruited at treatment centres than at the community level. The investigators should be aware, however, that comparisons based on hospital recruited cases and controls are subject to major biases especially when treatment seeking behaviour is not the same between the groups being compared as
---

	was already pointed out in the 1940's (See: Berkson J. Limitations of the application of fourfold table analysis to hospital data. Int J Epidemiol. 2014 Apr;43(2):511-5.). Clearly this will be a problem when comparing cases with well nourished controls, but it may also affect the comparison between oedematous and non oedematous malnutrition for which caretakers may have also different treatment seeking behaviour. Investigators should be aware of this problem and should be very cautious when interpreting their results. The associations they will find will be applicable only to inpatient settings and they will only help to generate hypothesis that will have to be tested independently by randomised trials before any firm conclusion can be drawn from them. 2) Anthropometric status of children In the section on study limitations, the authors mention that: “The clinical heterogeneity of the study participants, including comorbidities such as stunting and co-infections, may make it challenging to identify the specific causes of clinical outcomes”. This statement suggests that the associated stunting is a problem when interpreting the results of the study. This is not really true, and the associated stunting may even add value to the study. In most settings, children with SAM are not only wasted but also stunted and if this association is present in the study sample, this will increase the external validity of the study. Moreover, moderate wasting and stunting when present separately are associated with a moderate increase in the risk of death whereas when they are present together they are associated with a very high risk of death, suggesting a major interaction effect in their relationship with mortality. (See: McDonald CM, Olofin I, Flaxman S, Fawzi WW, Spiegelman D, Caulfield LE, Black RE, Ezzati M, Danaei G; Nutrition Impact Model Study. The effect of multiple anthropometric deficits on child mortality: meta-analysis of individual data in 10 prospective studies from developing countries. Am J Clin Nutr. 2013 Apr;97(4):896-901 and also: Myatt et al. Children who are both wasted and stunted (WaSt) are also underweight and have a high risk of death. 2017 Available at: https://www.enonline.net/attachments/2702/Myatt-et-al-WaSt-Final.pdf). The implication is that wasting and stunting should not be looked at separately as two independent risk factors and that the focus should be on their joint interactive effect. This can be achieved by mainly focusing the analysis on anthropometric indices influenced by both wasting and stunting (mid-upper arm circumference and weight-for-age). Minor remarks: The reference list should be thoroughly checked as there are duplicates and errors. Some duplicate references: 24 and 27; 28 and 39; 22 and 50; 23 and 60. 63 and 46. In reference 79 the first author is Schofield (check spelling). This reference is the same as reference 5.
--	---

REVIEWER	Hedwig DECONINCK
	Université catholique de Louvain Brussels
REVIEW RETURNED	16-Apr-2018

GENERAL COMMENTS	Please consider whether the following can be addressed (unless redundant): Uncomplicated versus complicated SAM Page 7/line 11: Two groups of SAM are identified in the introduction (uncomplicated and complicated SAM) indicating that children with uncomplicated SAM will be hospitalized, and with uncomplicated SAM not, but treated in the community. Page 11/line 15: For the recruitment and the numbers per study groups, the distinction of uncomplicated and complicated SAM is not made (and therefore not defined, and this may differ across the two countries) while SAM cases are stratified for non-oedematous and oedematous SAM. Not knowing the severity of SAM illness may induce loss of information or bias. OR, has the WHO 1999 treatment protocol been followed (as suggested on Page 16/lines 11-13, see comment below)? Page 12/line 23-31 Recruitment of control: The control should be well nourished and clinically well, but will have been admitted to the tertiary pediatric unit or as outpatient for a certain illness. Suggestion: It may be useful to identify the diagnosis for admission or reason for health seeking and examine whether there is no interference with the case definition. What is the definition of 'clinically well'? Which type of condition of clinically well children justifies admission to hospital or seeking healthcare? Page 15/line 14 Baseline data includes maternal and household characteristics Suggestion: This data may be very useful to understand vulnerability before and after treatment and it hopefully includes determinants of the socio-economic, health and sanitary environment (e.g., number of children in the household, number in line and birth spacing of the SAM child, feeding and care history, access to healthcare). Page 16/lines 11-13 The WHO 1999 treatment protocol will be followed Comment: 1) The WHO 1999 protocol has been improved with the WHO 2013 Update, and most national guidelines have taken these changes into account (e.g., distinguishing complicated and uncomplicated SAM); 2) Both Zambia and Zimbabwe have national SAM treatment protocols, and they may show small differences which may influence treatment outcomes (e.g., definitional of complicated SAM for inpatient admission, use of antibiotics and/or fluid management); have they been followed? It would be good to indicate more precisely what was used and applies and eventually, identify the differences between guidelines and take these into consideration. Or, does this suggest that there is no difference made between uncomplicated and complicated SAM, and that in both countries/hospitals all SAM are hospitalized? This is unclear (see the first comment). Page 16/lines 21-23 The WHO 2003 guidelines for HIV-naïve children Comment: The WHO 2013 Update covers guidance on start of ART in case of HIV with SAM. Page 16/lines 27-31 Discharge: the clinical team decides
--

	Comment: Indeed, the clinical team may decide when a child is ready for discharge from hospital to continue treatment in outpatient care until full recovery, but they adhere to certain criteria that are described in the national guidelines. It may be good to verify what the criteria are and whether they were adhered to. Page 16/lines 47-49 follow up at weeks 2-4-12-24 and 48 Post-discharge follow up may influence healthcare seeking and adherence behaviour. Would this information be captured? Also, post-discharge from hospital, depending on which and whether standardized criteria were applied, implies continuation of treatment of SAM in primary care on a (usual) weekly basis). Would this information be captured? Page 17/line 15 Children with relapsed malnutrition are provided with nutritional supplements according to local guidelines. If a child relapses (has a SAM condition as defined in the guidelines for SAM admission criteria, within an agreed period after discharge from treatment), then the child is readmitted for treatment, and not just 'given supplements'. The phrase is imprecise and should be: Children with relapsed malnutrition (within period x of discharge) or a new episode of malnutrition are admitted for treatment of SAM according to the national guidelines. It also contrasts with what is suggested on Page 19/lines 39-43: Children who are readmitted to one of the study sites with relapsed SAM have data collected during the new episode of hospitalisation.
--	--

REVIEWER	Moses Ngari KEMRI/wellcome Trust Research Programme, Kenya
REVIEW RETURNED	05-Jun-2018

GENERAL COMMENTS	The authors must be commended for setting a cohort study to investigate the long-term morbidity, growth and mortality comparing children with SAM and SAM-HIV during inpatient and post-discharge period. There are clear gaps in our knowledge of the pathogenies of SAM outcomes in long-term. The output of this cohort should not be limited to just conference or peer review publications but also generation of hypothesis that should be tested in clinical trials to reduce the approximately 5millions under-fives annual deaths. This is generally a well-designed cohort study, the authors should spare no effort to collect quality data and share the results promptly to influence current policies. Some comments to improve the protocol  -On page 6, lines 18-21 authors could also reference Ngari et.al Paediatric and Perinatal Epidemiology, 2017, 31, 233–242 that report high post-discharge mortality following treatment for pneumonia especially among young, malnourished and HIV infected children. -It worth stating despite children with SAM and SAM-HIV tend to have catch-up growth in weight-for-age and weight-for-length/height z-score, rarely they catch-up in length/height-for-age z-scores (page7 lines 14-19 (Berkley et.al Lancet Glob Health 2016)) -Please add that despite SAM and HIV sharing some features, while Co-trimoxazole as prophylaxis has been shown to reduce mortality by possibly reducing infections, it has not reduced
---

	mortality amongst children with SAM (reference: Chitu et.al Lancet 2004; 364: 1865–71 and Berkley et.al Lancet Glob Health 2016) and thus the need to understand this disparity (page 8, lines 29-42). -By defining SAM using MUAC<11.5cm for children≥6 months but only using weight-for-length<-3 or presence of edema amongst children<6months old might introduce some selection bias. The authors should consider some suggested MUAC cut-off for children<6 months. Oedema is very rare among this group, measuring length is challenging too and therefore the cohort might not pick proportion children in this age group. Mwangome et.al 2017 Am J Clin Nutr doi: 10.3945/ajcn.116.149815. has shown MUAC is most sensitive in identifying SAM compared to weight-for-length z-scores in this age group both during inpatient and one-year post-discharge and suggested a MUAC cut-off of 11cm (pages 9 and 10). -The authors should explain why they have excluded children<6months in the enteropathy substudy (page10, lines 19) -Despite stunting (length/height-for-age z-score<-2) being the most common form of childhood malnutrition affecting approximately 150million globally, previous studies suggest children recovering from SAM don't catch-up in length despite the rapid gain in weight. Is there an explanation why the authors did not consider this has a prime objective because such cohort would provide a rich mine to fill this gap?
--	--

REVIEWER	Stephanie Roll Charité Universitätsmedizin, Institute for Social Medicine, Epidemiology and Health Economics
REVIEW RETURNED	13-Jun-2018

GENERAL COMMENTS	BMJ Open Health Outcomes, Pathogenesis and Epidemiology of Severe Acute Malnutrition (HOPE-SAM): rationale and methods of a longitudinal observational study Manuscript ID bmjopen-2018-023077 The authors present the study design and methods of a prospective cohort study in children up to one year of age with Severe Acute Malnutrition (SAM) with and without HIV infection. The main aims of the study are to determine clinical outcomes of children with SAM with and without HIV, and to identify the risk factors for these outcomes. The manuscript is well written and illustrative. The study designs seem generally sound. The manuscript reports an ongoing study. The dates of the study are included in the manuscript. I would recommend the publication of the manuscript after revision. Please note: [ ] I reviewed this manuscript with emphasis on the statistical, epidemiological, and methodological aspects (not on clinical aspects). However, I am afraid that the assessment of some of the statistical analysis methods of the substudies on microbiota, metabolomics, and immunology (e.g. Principal coordinate analysis, multilevel simultaneous component analysis, orthogonal projections to latent structures models) are beyond my capabilities and I feel unable to provide a profound review of these methods.
--

During the revision process I would be willing to take direct questions from the authors regarding any of my concerns in order to facilitate the process.

Comments

1) Page 11 of 49 (Study design)

Line 19/29: Please describe the respective number of HIV-infected and HIV-uninfected children planned to be enrolled.

Please include the description of the microbiome, metabolome, and immunology substudies into this section. Please also please describe the selection processes for these substudies.

2) Page 12 of 49 (Eligibility for enteropathy substudy)

Line 19: Please be consistent with the labelling of the groups, i.e. it is not obvious what group A

(probably the combination of groups A-I and A-II?) and group C refer to.

Line 25: Please describe age bands here (instead of under table 1).

3) Page 12 of 49 (Table 1)

Line 53: Since the term 'MUAC' is not used in table 1, the explanation here is not necessary. However, please provide an explanation of MUAC at its first use in the text (Page 27 of 49, section 'Analysis').

4) Page 11 of 49 (Study design) or Page 12 of 49 (Eligibility for enteropathy substudy)

Please describe how the subgroup of 200 children is selected from the entire cohort.

5) Page 11 of 49 (Study design) and Page 46 of 49 (Figure)

Please describe consistently the recruitment of the control group of 200 healthy children for the enteropathy substudy: 'recruited from the same hospitals' vs. 'outpatients'/'study clinic/outpatients or ward'

6) Page 13 of 49 (Table 2)

There is no need to provide the names of the assessment forms, unless the assessment forms will be published alongside the manuscript (they were not part of the review documents). If assessment forms will be published, please use abbreviated names to improve the layout and readability of the table.

7) Page 19 of 49 (Substudies) and Page 46 of 49 (figure)

The hierarchy of the main study (cohort) and the substudies is a bit unclear. Is the design such that there is one main study (cohort) and one substudy (enteropathy) with 3 'sub-substudies' (microbiota, metabolomics, and immunology) to the enteropathy substudy, as outlined in figure 1? Or one main study (cohort) and 4 concurrent substudies (enteropathy, microbiota, metabolomics, and immunology) as described in section 'SUBSTUDIES' (Page 19 of 49)? Please describe more clearly.

8) Page 25 of 49 (Sample size)

Please provide an explanation of 'robust assessment of outcomes'. Is this term referring to precise estimators (i.e. narrow confidence intervals)? If so, of the entire cohort or of substudies?

Or is this term referring to any comparisons of groups, e.g. assessment of risk factors? Or has it other possible meanings? Which outcomes are referred to?

Could you describe how you come to the conclusion that 420 evaluable children are sufficient?

9) Page 26 of 49 (Sample size for Enteropathy substudy)

I don't think the formal alpha adjustment (Bonferroni correction) for two distinct group comparisons (HIV-SAM vs. SAM and HIV-SAM vs. HIV) regarding the LM ratios makes any sense. This is an exploratory observational study with many research questions, using many outcomes, many risk factors, several substudies (with many outcomes and influencing factors), and several group comparisons. Thus, to single out two comparisons of a substudy for confirmatory (?) assessments does not seem to be in line with the general aims of the study.

The same also applies to the sample size calculation of the inflammatory markers. Here, in addition, it seems that several outcomes ('inflammatory markers') will be compared (for HIV-SAM vs. SAM and HIV-SAM vs. HIV). Thus, there will be more comparisons performed than corrected for by the Bonferroni approach.

I understand that the sample size to be included into the study can hardly be changed at this point in time. The description of the sample size calculation could be revised, though.

Most importantly: since the sample size calculation should be in line with the statistical analysis, I strongly suggest not to use any alpha adjustments in the statistical analyses. Instead, please describe that all results will be interpreted exploratively.

10) Page 26 of 49, line 21-27 (Sample size for Enteropathy substudy)

Is this given reference the only source to assess the clinical relevance of LM-ratio? I.e. how will you be able to interpret the clinical relevance of your findings?

11) Page 26 of 49, line 37/39 (Sample size for Enteropathy substudy)

It is stated that 'Inclusion of healthy controls provides an indication of normal ranges in young African children. SAM'.

How will data from the healthy controls be analysed in the study? Will there be no formal analysis of differences between the 2 groups SAM vs. well-nourished children?

12) Anywhere between page 11 and page 26

I would suggest providing a specific section on study outcomes and risk factors (and their definitions).

13) Throughout entire manuscript

Please describe more clearly the groups to be compared. Throughout the manuscript, group comparisons are described as 'HIV-SAM compared with SAM', 'differences between SAM versus HIV-SAM', 'HIV-SAM versus SAM', 'children with SAM compared to HIV-SAM' etc.

	While it is clear that ‘HIV-SAM’ denotes HIV-infected children with SAM, the notation for the ‘SAM’ children is not clear. Who exactly is included in the ‘SAM’ group? All children with SAM (regardless of HIV status)? Or only children with SAM but without HIV? If the latter is true, maybe the term ‘nonHIV-SAM’ would be useful to denote this group. [ ] Similarly, in the comparison described as ‘HIV-SAM versus HIV’ (Page 26 of 49, line 17) it is unclear which children belong to the HIV-group (well-nourished children with HIV?). 14) Page 27 of 49 (ANALYSIS Observational Cohort) [ ] Line 45/46 and 53: Please describe more clearly what is meant by ‘will be determined for each group’. Will the two groups (HIV-SAM and non-HIV-SAM) be included in one model (together with the risk factors)? If so, will interaction terms be used to assess if associations between risk factors and outcomes are different between the two groups? Or will the analysis of associations between risk factors and outcomes be performed separately for each of the two groups? [ ] Line 47 and other appearances throughout the manuscript: Please use the term ‘multivariable analysis’ instead of ‘multivariate analysis’ when single outcomes are analysed. 15) Page 29 of 49 (ANALYSIS Enteropathy substudy) [ ] Line 3: Instead of simple descriptive statistics, I would suggest to use appropriate models to allow for the adjustment of possible relevant differences in baseline characteristics. [ ] Line 11: Please describe which interaction terms will be included into the models (i.e. interactions between which factors?). [ ] Line 23: Please describe which groups and which group comparisons will be performed. [ ] Line 29: Instead of Kaplan-Meier analyses and log-rank tests, I would suggest using Cox models to allow for the adjustment of possible relevant baseline differences between the comparison groups. 16) Page 47 of 49 (Supplementary Table 1) [ ] I would suggest considering to replace the last column (time-points) with 5 columns (e.g. labelled: B, D, 12, 24, 48) and to use crosses (x) as indications for the respectively used assessments time-points to improve readability. [ ] Line 57: Please provide the location of the Blizzard Institute 17) Page 49 of 49 (Supplementary table 2) [ ] Column ‘Study groups’: please provide a description of the groups (A, B, C, D) below the table (instead of giving a reference to table 1).
--	--

VERSION 1 – AUTHOR RESPONSE

Reviewer: 1 Reviewer Name: André BRIEND

Children with severe acute malnutrition (SAM) have a high mortality even when treated with currently recommended WHO protocol. The reasons for this are not clear, and in depth studies examining

possible mechanisms are most welcome. Following the development of ready-to-use therapeutic foods (RUTF) research on SAM management have focused over the last 15 years on how to simplify treatment and make it more accessible to large number of children. This was a priority, but improving treatment of complicated cases is also needed and this requires a better understanding of current treatment limitations. This proposal addresses precisely this important issue. It will provide invaluable information on how to improve survival of these high risk children.

This protocol is a result of an impressive team work involving many high level specialists in different areas and at this stage, there is not much to comment about this ongoing study. Two points, however, may be highlighted.

1) Interpreting the results.

The whole protocol is based on the comparison of children with SAM, HIV-infected or HIV uninfected with and without oedema, with controls taken among children receiving inpatient or outpatient care at the study site. The rationale for this is clear, patients but also controls are more easily recruited at treatment centres than at the community level. The investigators should be aware, however, that comparisons based on hospital recruited cases and controls are subject to major biases especially when treatment seeking behaviour is not the same between the groups being compared as was already pointed out in the 1940's (See: Berkson J. Limitations of the application of fourfold table analysis to hospital data. *Int J Epidemiol.* 2014 Apr;43(2):511-5.). Clearly this will be a problem when comparing cases with well nourished controls, but it may also affect the comparison between oedematous and non oedematous malnutrition for which caretakers may have also different treatment seeking behaviour. Investigators should be aware of this problem and should be very cautious when interpreting their results. The associations they will find will be applicable only to inpatient settings and they will only help to generate hypothesis that will have to be tested independently by randomised trials before any firm conclusion can be drawn from them.

Response: We thank the Reviewer for this important comment and we acknowledge the challenge of bias in selecting controls for case-control studies. To mitigate this, we are also enrolling some controls from the communities in which the cases with SAM are drawn, including healthy siblings of cases, in addition to hospital-based controls from outpatient clinics or other wards. This was omitted from the first version of the manuscript and has been updated. We accept that the case-control study is predominantly hypothesis-generating, and aims to provide normative values for many of the biomarkers being measured in cases with SAM, and we will be careful in our interpretation of findings to point out the challenges inherent in case-control selection.

We have updated the "Eligibility for enteropathy substudy" section to read (Page 11, line 23 onwards): "*Well-nourished controls*: Controls are children drawn from the same hospitals and communities as cases with SAM (including well-nourished sibling controls), who are aged 6-59 months (matched to enteropathy substudy children within age bands), well-nourished (weight-forheight Z-score >-1) and clinically well (no acute illness or current infections) with known HIV status. Controls are categorized into two groups: wellnourished HIV-positive (Group B) and well-nourished HIV-negative (Group D), as shown in Table 1."

We have also added an additional bullet point to the 'limitations' section to reflect this point. "Potential bias in recruiting well-nourished controls only from hospitals will be reduced by inclusion of community-based controls, including well-nourished siblings of children with SAM."

2) Anthropometric status of children

In the section on study limitations, the authors mention that:

“The clinical heterogeneity of the study participants, including comorbidities such as stunting and co-infections, may make it challenging to identify the specific causes of clinical outcomes”.

This statement suggests that the associated stunting is a problem when interpreting the results of the study. This is not really true, and the associated stunting may even add value to the study. In most settings, children with SAM are not only wasted but also stunted and if this association is present in the study sample, this will increase the external validity of the study. Moreover, moderate wasting and stunting when present separately are associated with a moderate increase in the risk of death whereas when they are present together they are associated with a very high risk of death, suggesting a major interaction effect in their relationship with mortality. (See: McDonald CM, Olofin I, Flaxman S, Fawzi WW, Spiegelman D, Caulfield LE, Black RE, Ezzati M, Danaei G; Nutrition Impact Model Study. The effect of multiple anthropometric deficits on child mortality: meta-analysis of individual data in 10 prospective studies from developing countries. *Am J Clin Nutr*. 2013 Apr;97(4):896901 and also: Myatt et al. Children who are both wasted and stunted (WaSt) are also underweight and have a high risk of death. 2017 Available at: <https://www.ennonline.net/attachments/2702/Myatt-et-al-WaSt-Final.pdf>). The implication is that wasting and stunting should not be looked at separately as two independent risk factors and that the focus should be on their joint interactive effect. This can be achieved by mainly focusing the analysis on anthropometric indices influenced by both wasting and stunting (mid-upper arm circumference and weight-for-age).

Response: We agree that stunting frequently overlaps with wasting and compounds mortality risk, and that excluding children with stunting would not be appropriate. The limitation we were pointing out is that separating out the impact of stunting from the impact of SAM is difficult due to the overlapping and interacting nature of these anthropometric defects, but we accept that this is not a limitation per se, but a potential strength. We have therefore removed ‘stunting’ from this bullet point, which now reads: “The clinical heterogeneity of the study participants, including comorbidities such as co-infections, may make it challenging to identify the specific causes of clinical outcomes. However, the embedded sub-studies will enable multiple pathways to be explored within the same cohort.”

Minor remarks:

The reference list should be thoroughly checked as there are duplicates and errors. Some duplicate references: 24 and 27; 28 and 39; 22 and 50; 23 and 60. 63 and 46. In reference 79 the first author is Schofield (check spelling). This reference is the same as reference 5.

Response: We have checked and updated all references in the revised manuscript.

Reviewer: 2 Reviewer Name: Hedwig DECONINCK

Well developed protocol for a great study.

Please consider whether the following can be addressed (unless redundant):

Uncomplicated versus complicated SAM Page 7/line 11: Two groups of SAM are identified in the introduction (uncomplicated and complicated SAM) indicating that children with uncomplicated SAM will be hospitalized, and with uncomplicated SAM not, but treated in the community. Page 11/line 15: For the recruitment and the numbers per study groups, the distinction of uncomplicated and complicated SAM is not made (and therefore not defined, and this may differ across the two countries) while SAM cases are stratified for non-oedematous and oedematous SAM. Not knowing the severity of SAM illness may induce loss of information or bias. OR, has the WHO 1999 treatment protocol been followed (as suggested on Page 16/lines 1113, see comment below)?

Response: We apologise for the ambiguity introduced in the original manuscript. This study only enrolls children with complicated SAM, because all are hospitalized cases. We did not specify in the inclusion criteria the reason why each child was diagnosed with complicated SAM; rather, if a child was hospitalized and met anthropometric or oedema criteria for SAM by WHO guidelines, they were eligible. We have clarified this in the screening section (page 10, lines 8-13): "All new admissions aged 0-59 months are screened for

SAM, which is defined according to WHO criteria as any of: weight-for-height Z-score (WHZ) <-3, mid-upper arm circumference <115 mm (if aged 6-59mo) and/or bilateral pitting oedema. All children with SAM are recruited from hospital and this study therefore focuses on complicated SAM; children with uncomplicated SAM will not be enrolled."

Page 12/line 23-31 Recruitment of control: The control should be well nourished and clinically well, but will have been admitted to the tertiary pediatric unit or as outpatient for a certain illness.

Suggestion: It may be useful to identify the diagnosis for admission or reason for health seeking and examine whether there is no interference with the case definition. What is the definition of 'clinically well'? Which type of condition of clinically well children justifies admission to hospital or seeking healthcare?

Response: We define 'clinically well' in the study protocol as having no acute illness or current infections. In general, controls are well-nourished children attending outpatient clinics (including the HIV clinic) for follow-up after a recent ward admission. However, as stated above, we are also enrolling some healthy community controls, including siblings of cases with SAM. We undertake an extensive baseline interview in which we capture details of recent illness and underlying medical conditions so we will be able to characterize the controls well.

Page 15/line 14 Baseline data includes maternal and household characteristics

Suggestion: This data may be very useful to understand vulnerability before and after treatment and it hopefully includes determinants of the socio-economic, health and sanitary environment (e.g., number of children in the household, number in line and birth spacing of the SAM child, feeding and care history, access to healthcare).

Response: We capture data on these variables in our baseline form, recognizing that the home and family environment is a critical determinant of malnutrition and of long-term outcomes following hospitalization. As explained in the revised manuscript, these variables will be used to evaluate baseline risk factors and to adjust for confounding (Page 27, lines 1-5): "Data on potential confounders are collected at baseline, discharge and during the follow-up period, including child feeding practices, household socioeconomic status (defined by household income and cooking method), maternal employment and education, and household factors such as water, sanitation and hygiene practices, availability of electricity, location (rural, peri-urban or urban) and household size."

Page 16/lines 11-13 The WHO 1999 treatment protocol will be followed

Comment: 1) The WHO 1999 protocol has been improved with the WHO 2013 Update, and most national guidelines have taken these changes into account (e.g., distinguishing complicated and uncomplicated SAM); 2) Both Zambia and Zimbabwe have national SAM treatment protocols, and they may show small differences which may influence treatment outcomes (e.g., definitional of complicated SAM for inpatient admission, use of antibiotics and/or fluid management); have they been followed? It would be good to indicate more precisely what was used and applied and eventually, identify the differences between guidelines and take these into consideration. Or, does this suggest that there is no difference made between uncomplicated and complicated SAM, and that in both countries/hospitals all SAM are hospitalized? This is unclear (see the first comment).

Response: As discussed above, all children in this study have complicated SAM. The reviewer is correct that each site follows national guidelines, which are based on WHO 1999 (and 2013 update) guidelines, but do have some differences. We have been very careful to capture extensive daily data on management of each child in hospital (including use of fluids, blood transfusions, antibiotics and monitoring undertaken) so that we can evaluate whether differences in management across sites and between children contributes to outcomes. We have included reference to the 2013 WHO guidelines now, wherever the original 1999 guideline are cited, and added a line on this issue in the revised manuscript, which now reads (page 14, line 1924): "In addition, the HOPE-SAM study clinician at each hospital site collects daily data until discharge on clinical parameters (including daily examination), resolution of acute infections, nutritional recovery (loss of oedema, restoration of appetite, weight gain), and treatment/nutritional supplements received; this will allow us to evaluate differences in management between countries." Page 16/lines 21-23 The WHO 2003 guidelines for HIV-naïve children

Comment: The WHO 2013 Update covers guidance on start of ART in case of HIV with SAM.

Response: Thank you for pointing this out. We made an error here and cited the 2003 HIV guidelines instead of the current WHO guidelines on when to start ART. This has been corrected and we have also included the 2013 SAM guidelines as suggested. ART is started according to national guidelines, which are based on all these WHO recommendations, so we have updated the whole sentence to read (page 14, lines 24-26): "Children with HIV-SAM who are ART-naïve start ART according to national guidelines, which are based on WHO recommendations."

Page 16/lines 27-31 Discharge: the clinical team decides

Comment: Indeed, the clinical team may decide when a child is ready for discharge from hospital to continue treatment in outpatient care until full recovery, but they adhere to certain criteria that are described in the national guidelines. It may be good to verify what the criteria are and whether they were adhered to.

Response: Discharge criteria are now based on clinical judgement, with no need to reach specific weight-for-height or weight gain criteria in either country. In general, once oedema and complications are resolving, and children have a good appetite and are clinically well and alert, they are discharged. We have added the following to the discharge section (Page 15, lines 1-3): "*Discharge*: The clinical team decides when the child is ready to be discharged, which is generally when their medical complications are resolving and the child has a good appetite and is clinically well and alert."

Page 16/lines 47-49 follow up at weeks 2-4-12-24 and 48

Post-discharge follow up may influence healthcare seeking and adherence behaviour. Would this information be captured? Also, post-discharge from hospital, depending on which and whether standardized criteria were applied, implies continuation of treatment of SAM in primary care on a (usual) weekly basis). Would this information be captured?

Response: Yes, we capture data on use of RUTF after discharge, illness episodes and healthcare-seeking behavior including clinic attendances. We agree that all these may be important determinants of outcomes.

Page 17/line 15 Children with relapsed malnutrition are provided with nutritional supplements according to local guidelines. If a child relapses (has a SAM condition as defined in the guidelines for SAM admission criteria, within an agreed period after discharge from treatment), then the child is readmitted for treatment, and not just 'given supplements'. The phrase is imprecise and should be: Children with relapsed malnutrition (within period x of discharge) or a new episode of malnutrition are admitted for treatment of SAM according to the national guidelines. It also contrasts with what is suggested on Page 19/lines 39-43: Children who are readmitted to one of the study sites with relapsed SAM have data collected during the new episode of hospitalisation.

Response: The management of relapse depends on the clinical status of the child. If the child has moderate acute malnutrition, they may just receive nutritional supplements (e.g. corn-soya blend); if uncomplicated SAM they will receive RUTF and if complicated SAM they will be readmitted to hospital. There are differences in criteria across countries and we do not have sufficient space in this paper to describe all these scenarios in detail. We capture data on all these possibilities in our follow-up form; we only collect hospitalization data if the child relapses with complicated SAM. We have changed the sentence to read: "Children with relapsed malnutrition are provided with nutritional supplements or RUTF according to local guidelines, or readmitted to hospital if they develop complicated SAM."

Reviewer: 3 Reviewer Name: Moses Ngari

The authors must be commended for setting a cohort study to investigate the longterm morbidity, growth and mortality comparing children with SAM and SAM-HIV during inpatient and post-discharge period. There are clear gaps in our knowledge of the pathogenies of SAM outcomes in long-term. The output of this cohort should not be limited to just conference or peer review publications but also generation of hypothesis that should be tested in clinical trials to reduce the approximately 5millions under-fives annual deaths. This is generally a well-designed cohort study, the authors should spare no effort to collect quality data and share the results promptly to influence current policies.

Response: Thank you for these comments. We wholeheartedly agree that the findings of this study should not be limited to publication; our goal is absolutely to define new intervention approaches. To reflect this, we have rephrased the last line of the paper (Page 33, lines 26-28), which now reads: "Our ultimate goal is to utilise the findings generated in this study to inform new intervention approaches that can be evaluated in clinical trials to improve outcomes among children with SAM."

Some comments to improve the protocol

-On page 6, lines 18-21 authors could also reference Ngari et.al Paediatric and Perinatal Epidemiology, 2017, 31, 233–242 that report high post-discharge mortality following treatment for pneumonia especially among young, malnourished and HIV infected children.

Response: We have added this reference as suggested.

-It worth stating despite children with SAM and SAM-HIV tend to have catch-up growth in weight-for-age and weight-for-length/height z-score, rarely they catch-up in length/height-for-age z-scores (page7 lines 14-19 (Berkley et.al Lancet Glob Health 2016))

Response: Thank you for highlighting this important point. We have amended this sentence to read: "Children with HIV-SAM appear to have potential for catch-up growth in weight-for-age and/or weight-for-height, which have been shown to normalise with treatment even prior to widespread availability of ART; by contrast, height-for-age shows less potential for catch-up growth."

-Please add that despite SAM and HIV sharing some features, while Co-trimoxazole as prophylaxis has been shown to reduce mortality by possibly reducing infections, it has not reduced mortality amongst children with SAM (reference: Chitu et.al Lancet 2004; 364: 1865–71 and Berkley et.al Lancet Glob Health 2016) and thus the need to understand this disparity (page 8, lines 29-42).

Response: Thank you for this suggestion. We agree that cotrimoxazole has generally shown benefits in HIV-infected populations (e.g. the CHAP trial in 2004) and less clear (if any) benefits in HIV-uninfected populations. However, we are not aware of studies that have specifically evaluated the impact of cotrimoxazole in the setting of HIV-SAM. The CHAP trial enrolled hospitalized children in the pre-ART era and did not specifically evaluate outcomes in the subgroup with SAM, whilst the Kenyan

CTX trial (Berkley et al) excluded children with HIV. Cotrimoxazole is now recommended long-term for all children with HIV, regardless of disease stage. We are therefore not keen to elaborate further on this point in the manuscript because, in the absence of data, we do not feel able to make the point that it clearly differs in activity between HIV groups in the setting of SAM.

-By defining SAM using MUAC<11.5cm for children≥6 months but only using weight-for-length<-3 or presence of edema amongst children<6months old might introduce some selection bias. The authors should consider some suggested MUAC cut-off for children<6 months. Oedema is very rare among this group, measuring length is challenging too and therefore the cohort might not pick proportion children in this age group. Mwangome et.al 2017 Am J Clin Nutr doi: 10.3945/ajcn.116.149815. has shown MUAC is most sensitive in identifying SAM compared to weight-for-length zscores in this age group both during inpatient and one-year post-discharge and suggested a MUAC cut-off of 11cm (pages 9 and 10).

Response: Thank you for this suggestion. We do intend to use the Mwangome et al data to explore the MUAC criteria for SAM in children under 6 months of age, although we utilised WHO criteria for diagnosis of SAM in this age group as an inclusion criterion to the study for consistency. Our analyses using MUAC will therefore be exploratory in the enrolled population, but we expect them to shed light on this important issue of how to define SAM <6mo of age. We have revised the analysis section to reflect how we may use the

Mwangome et al data to evaluate this issue in infants under 6 months (Page 27, lines 21-27): “We will evaluate the ability of mid-upper arm circumference (MUAC) at discharge to predict long-term outcomes using receiver-operatorcharacteristic (ROC) analysis, in the whole cohort and within the subgroups of HIV-SAM and HIV-negative children with SAM. We will then evaluate whether addition of other variables improves the predictive capacity of MUAC (using WHO criteria in those >6 months old, and published data for children <6 months) for each group...”

-The authors should explain why they have excluded children<6months in the enteropathy substudy (page10, lines 19)

Response: We apologise for not making this clear in the original submission. We chose to exclude children <6mo of age from the enteropathy substudy because of the requirement for a lactulose-mannitol test, which would interrupt exclusive breastfeeding at this age; since it is a research test rather than a clinical test, we did not feel it was justifiable. We have updated the footnote in Table 1 to read: “Note that children below 6 months of age are excluded from the enteropathy substudy to avoid interrupting exclusive breastfeeding during the lactulose-mannitol test.”

-Despite stunting (length/height-for-age z-score<-2) being the most common form of childhood malnutrition affecting approximately 150million globally, previous studies suggest children recovering from SAM don't catch-up in length despite the rapid gain in weight. Is there an explanation why the authors did not consider this has a prime objective because such cohort would provide a rich mine to fill this gap?

Response: Although this is not a primary objective of the study, which focuses predominantly on relapse of SAM, we will certainly be able to explore this during follow-up, since LAZ will be calculated at every visit. We agree that this is an important question to be addressed and thank the reviewer for the suggestion.

Reviewer: 4 Reviewer Name: Stephanie Roll

The authors present the study design and methods of a prospective cohort study in children up to one year of age with Severe Acute Malnutrition (SAM) with and without HIV infection. The main aims of

the study are to determine clinical outcomes of children with SAM with and without HIV, and to identify the risk factors for these outcomes.

The manuscript is well written and illustrative. The study designs seem generally sound. The manuscript reports an ongoing study. The dates of the study are included in the manuscript. I would recommend the publication of the manuscript after revision.

Please note:

- I reviewed this manuscript with emphasis on the statistical, epidemiological, and methodological aspects (not on clinical aspects). However, I am afraid that the assessment of some of the statistical analysis methods of the substudies on microbiota, metabolomics, and immunology (e.g. Principal coordinate analysis, multilevel simultaneous component analysis, orthogonal projections to latent structures models) are beyond my capabilities and I feel unable to provide a profound review of these methods.
- During the revision process I would be willing to take direct questions from the authors regarding any of my concerns in order to facilitate the process. Comments

. 1) Page 11 of 49 (Study design)

- Line 19/29: Please describe the respective number of HIV-infected and HIV-uninfected children planned to be enrolled.

Response: For the observational study, we enrolled all eligible children with SAM provided caregivers were willing to consent. We did not have a specific target for numbers of HIV-infected versus HIV-uninfected children, but a total target of 600-800 children regardless of HIV status. For the enteropathy substudy, we had specific targets split by HIV status, as shown in Table 1, which was based on our sample size required to detect important differences in enteropathy biomarkers between groups.

- Please include the description of the microbiome, metabolome, and immunology substudies into this section. Please also please describe the selection processes for these substudies.

- . Response: Thank you for this suggestion. We have extensively rewritten the early sections of the paper to clarify the sub-studies. We introduce them in the Study Design section and we have revised Figure 1 to more clearly show how the 4 sub-studies relate to each other, and which children are included in each. We give the eligibility criteria for each sub-study in the Recruitment section. We then describe the substudies in more detail later, since that requires knowledge of the followup processes in HOPE SAM. Overall, we feel this has clarified the overall design of HOPE SAM and how the sub-studies fit in, and thank the Reviewer for the suggestion.

. 2) Page 12 of 49 (Eligibility for enteropathy substudy)

- Line 19: Please be consistent with the labelling of the groups, i.e. it is not obvious what group A (probably the combination of groups A-I and A-II?) and group C refer to.
- Line 25: Please describe age bands here (instead of under table 1).

Response: We apologise for the ambiguity in labeling; the Reviewer is correct that group A comprises A-I and A-II; however, we have now outlined this more clearly, and included the age bands in this section, which now reads:

“Eligibility for enteropathy substudy: Children with SAM aged 6-59 months with a nasogastric tube in place (or due to be placed) are categorized into 4 groups (HIV-positive oedematous (Group A-I); HIV-

positive non-oedematous (Group A-II); HIV-negative oedematous (Group C-I) and HIV-negative nonoedematous (Group C-II), as shown in Table 1. Children meeting eligibility criteria will be enrolled throughout the study recruitment period until sufficient specimens have been collected from the groups shown in Table 1. Children in the enteropathy substudy are stratified into age bands (6-11 months; 12-23 months and 24-59 months) to enable age-matching of well-nourished controls. Children with underlying chronic gastrointestinal disease or a known malignancy are ineligible.”

. 3) Page 12 of 49 (Table 1)

- Line 53: Since the term 'MUAC' is not used in table 1, the explanation here is not necessary.

However, please provide an explanation of MUAC at its first use in the text (Page 27 of 49, section 'Analysis').

Response: We have made these changes as suggested.

4) Page 11 of 49 (Study design) or Page 12 of 49 (Eligibility for enteropathy substudy) - Please describe how the subgroup of 200 children is selected from the entire cohort.

Response: This has now been clarified in the eligibility section (page 10, lines 24-26): “Children meeting eligibility criteria will be enrolled throughout the study recruitment period until sufficient specimens have been collected from the groups shown in Table 1.”

5) Page 11 of 49 (Study design) and Page 46 of 49 (Figure) - Please describe consistently the recruitment of the control group of 200 healthy children for the enteropathy substudy: ‘recruited from the same hospitals’ vs. ‘outpatients’/ ‘study clinic/outpatients or ward’

Response: Please also see the response also to Reviewer 1 on this point about the population from which controls are drawn. Since the initial drafting of this paper we have extended our enrolment strategy; as well as recruiting children from inpatient and outpatient facilities at the hospital sites, we are also enrolling children (including well-nourished siblings) from the same communities as children with SAM, to reduce bias regarding health-seeking behavior and increase recruitment. We have clarified this throughout the text and ensured consistent language; for example, use of ‘well-nourished’ throughout instead of healthy, and consistent use of ‘hospital and community controls’.

Under eligibility criteria, the description now reads (Page 11, line 23 onwards): “*Well-nourished controls*: Controls are children drawn from the same hospitals and communities as cases with SAM (including well-nourished sibling controls), who are aged 6-59 months (matched to enteropathy substudy children within age bands), well-nourished (weight-for-height Z-score >-1) and clinically well (no acute illness or current infections) with known HIV status. Controls are categorized into two groups: well-nourished HIV-positive (Group B) and well-nourished HIV-negative (Group D), as shown in Table 1. Children with underlying chronic gastrointestinal disease or a known malignancy are ineligible.”

6) Page 13 of 49 (Table 2) - There is no need to provide the names of the assessment forms, unless the assessment forms will be published alongside the manuscript (they were not part of the review documents). If assessment forms will be published, please use abbreviated names to improve the layout and readability of the table.

Response: The form names have been removed.

7) Page 19 of 49 (Substudies) and Page 46 of 49 (figure) - The hierarchy of the main study (cohort) and the substudies is a bit unclear. Is the design such that there is one main study (cohort) and one substudy (enteropathy) with 3 ‘sub-substudies’ (microbiota, metabolomics, and immunology) to the enteropathy substudy, as outlined in figure 1? Or one main study (cohort) and 4 concurrent

substudies (enteropathy, microbiota, metabolomics, and immunology) as described in section 'SUBSTUDIES' (Page 19 of 49)? Please describe more clearly.

Response: We apologise for this ambiguity. We think the revisions to the manuscript (outlined in point 1, above) have clarified this now, particularly the revised Figure 1.

8) Page 25 of 49 (Sample size)

- Please provide an explanation of 'robust assessment of outcomes'. Is this term referring to precise estimators (i.e. narrow confidence intervals)? If so, of the entire cohort or of substudies? Or is this term referring to any comparisons of groups, e.g. assessment of risk factors? Or has it other possible meanings? Which outcomes are referred to?

Response: We apologise for the unclear use of this term, which was not meant to refer to precision and has been removed; we have also clarified the outcomes referred to. We have therefore amended this sentence to read (Page 24, lines 2-5): "The observational cohort will recruit as many children with SAM as possible during the period of enrolment (July 2016 to March 2018), estimated at 600-800 children (capped at 800 maximum), to assess clinical and nutritional outcomes among HIV-positive and HIV-negative children hospitalised with SAM."

- Could you describe how you come to the conclusion that 420 evaluable children are sufficient?

Response: We apologise that the sample size calculation for the observational study had not been included in the original submission. This has now been added (Page 24, lines 6-11): "Assuming mortality of 15%, overall loss to followup of 15% and recruitment target of 800 children, there would be 560 evaluable children at 48 weeks, of whom 224 would have HIV-SAM based on an estimated inpatient HIV prevalence of 40%. This will provide >80% power to detect absolute differences of 17% in binary outcomes between HIV-SAM and HIVnegative children with SAM, and of 0.33 times the standard deviation in continuous outcomes."

9) Page 26 of 49 (Sample size for Enteropathy substudy)

- I don't think the formal alpha adjustment (Bonferroni correction) for two distinct group comparisons (HIV-SAM vs. SAM and HIV-SAM vs. HIV) regarding the LM ratios makes any sense. This is an exploratory observational study with many research questions, using many outcomes, many risk factors, several substudies (with many outcomes and influencing factors), and several group comparisons. Thus, to single out two comparisons of a substudy for confirmatory (?) assessments does not seem to be in line with the general aims of the study.
- The same also applies to the sample size calculation of the inflammatory markers. Here, in addition, it seems that several outcomes ('inflammatory markers') will be compared (for HIV-SAM vs. SAM and HIV-SAM vs. HIV). Thus, there will be more comparisons performed than corrected for by the Bonferroni approach.

Response: We agree that p-values should not be artificially adjusted ("alpha adjustment") when reporting results and would not aim to do that, but rather interpret the p-value as indicating the strength of evidence supporting any association. All results will be interpreted exploratively, since we agree with the reviewer that this is an exploratory observational study with many research questions using many outcomes, many risk factors etc. We have therefore added a sentence on these two points to the revised manuscript at the start of the Analysis section (Page 27, lines 8-13): "All analyses will be interpreted exploratively since HOPE-SAM is an observational study with multiple risk factors,

outcomes and substudies. For all analyses, P values will not be artificially adjusted, but interpreted as exploring the strength of evidence supporting any association. The only exception is the use of approaches to minimise false discovery when analysing high-dimensional data from the microbiota and metabolomics substudies, as described.”

However, we do differ in our view about the best way to determine and present the sample size for this study. Specifically, we have pre-specified one primary outcome measure for the enteropathy substudy (mean LM ratio), and effectively we have three main groups we wish to compare across this outcome (HIVSAM, HIV-negative children with SAM, and well-nourished HIV positive children) in two pairwise comparisons (HIV-SAM vs SAM, HIV-SAM vs HIV). If we were running an RCT with three randomised groups, two pairwise comparisons vs a control, and one primary outcome measure, it would be expected that the sample size calculation would include alpha adjustment as we have done here (regardless of how p-values are reported, see comment on strength of evidence above), to ensure that the trial has adequate size to allow for chance differences between the observed and true control group event rate. We do feel that we should follow the same procedure for an observational study, given that we have pre-specified a single primary outcome measure, and two primary pairwise comparisons with one group (HIV-SAM) in common. We would further note that this sample size calculation was accepted by all the peer reviewers for the MRC panel that funded the study, and by the ethical committees that approved the protocol. Further we feel that the sample size calculation reported in this paper should not differ from that in the ethically approved protocol and the awarded grant.

- I understand that the sample size to be included into the study can hardly be changed at this point in time. The description of the sample size calculation could be revised, though.
2/4

Response: As described above, the sample size for the observational study outcomes have now been added, and the enteropathy substudy sample size calculation is justified above.

- Most importantly: since the sample size calculation should be in line with the statistical analysis, I strongly suggest not to use any alpha adjustments in the statistical analyses. Instead, please describe that all results will be interpreted exploratively.

Response: We agree entirely, and have addresses this point above, with additions to the revised manuscript to explain this (page 27, lines 8-13).

10) Page 26 of 49, line 21-27 (Sample size for Enteropathy substudy) - Is this given reference the only source to assess the clinical relevance of LM-ratio? I.e. how will you be able to interpret the clinical relevance of your findings?

Response: This was the seminal Gambian study available when we designed the study which informed the sample size. However, there are many more recent studies that will enable clinical comparison of our cohort with others. In particular, we carefully aligned our lactulose-mannitol testing protocol with that developed by the multi-country Mal-ED study. Mal-ED has now published extensive data on LM ratios from 8 countries (e.g. Lee GO, Am J Trop Med Hyg 2017; Kosek MN, J Pediatr Gastroenterol Nutr 2017) which we will be able to reference for interpretation of our own data.

11) Page 26 of 49, line 37/39 (Sample size for Enteropathy substudy) - It is stated that 'Inclusion of healthy controls provides an indication of normal ranges in young African children. SAM'. How will data from the healthy controls be analysed in the study? Will there be no formal analysis of differences between the 2 groups SAM vs. well-nourished children?

Response: Yes, there will be formal comparison of biomarker data with wellnourished controls, as explained in the response above about our reasons for alpha adjustment in calculating the sample size. First, biomarkers measured in HIV-positive children with SAM will be compared with HIV-negative children with SAM (to assess the impact of HIV on biomarkers, since both groups have SAM) and, second, with the well-nourished HIV-positive group (to assess the impact of SAM, since both groups have HIV). The additional benefit of the healthy controls, which we were pointing out in this sentence, is to provide an understanding of the normal range of biomarkers in this setting – data that are lacking for African children.

12) Anywhere between page 11 and page 26 - I would suggest providing a specific section on study outcomes and risk factors (and their definitions).

Response: Thank you. We have added this section as suggested, on page 2627, which outlines risk factors, outcomes and covariates.

13) Throughout entire manuscript

- Please describe more clearly the groups to be compared. Throughout the manuscript, group comparisons are described as 'HIV-SAM compared with SAM', 'differences between SAM versus HIV- SAM', 'HIV-SAM versus SAM', 'children with SAM compared to HIV-SAM' etc. While it is clear that 'HIVSAM' denotes HIV-infected children with SAM, the notation for the 'SAM'- children is not clear. Who exactly is included in the 'SAM' group? All children with SAM (regardless of HIV status)? Or only children with SAM but without HIV? If the latter is true, maybe the term 'non- HIV-SAM' would be useful to denote this group.
- Similarly, in the comparison described as 'HIV-SAM versus HIV' (Page 26 of 49, line 17) it is unclear which children belong to the HIV-group (wellnourished children with HIV?).

Response: We apologise for the ambiguity in this terminology and recognize now that it confusing to read. We have clarified the whole manuscript by being consistent in the terms we use to describe groups. In the revised manuscript, the term SAM refers to all children with severe acute malnutrition; if the HIV status of the group is relevant, we use the following terms: "HIV-negative children with SAM", and either "HIV-positive children with SAM" or "HIV-SAM". We have clarified this early in the study design section (page 9, lines 14-17): "Throughout this paper, 'SAM' refers to all children, regardless of HIV status; where analyses specifically compare children by HIV status, groups are identified as HIV-positive children with SAM (or HIV-SAM) and HIV-negative children with SAM."

14) Page 27 of 49 (ANALYSIS Observational Cohort)

- Line 45/46 and 53: Please describe more clearly what is meant by 'will be determined for each group'. Will the two groups (HIV-SAM and non-HIVSAM) be included in one model (together with the risk factors)? If so, will interaction terms be used to assess if associations between risk factors and outcomes are different between the two groups? Or will the analysis of associations between risk factors and outcomes be performed separately for each of the two groups?

Response: We apologise for not being clear about how we plan to conduct this analysis. The reviewer is correct that we plan to include the two groups (HIVpositive children with SAM and HIV-negative children with SAM) in one model together with the risk factors, and use interaction tests to investigate whether associations between risk factors and outcomes differ between the two groups. We have added this to the manuscript (Page 27, line 28): "HIV-positive children with SAM and HIV-negative children with SAM will be included in one model together with the risk factors, and interaction tests will be used to investigate whether associations between risk factors and outcomes differ between the two groups of children."

- Line 47 and other appearances throughout the manuscript: Please use the term 'multivariable analysis' instead of 'multivariate analysis' when single outcomes are analysed. 3/4

Response: We have made these changes throughout

15) Page 29 of 49 (ANALYSIS Enteropathy substudy)

- Line 3: Instead of simple descriptive statistics, I would suggest to use appropriate models to allow for the adjustment of possible relevant differences in baseline characteristics.

Response: Simple descriptive statistics will be used in addition to regression models and multilevel models, as explained later in this section.

- Line 11: Please describe which interaction terms will be included into the models (i.e. interactions between which factors?).

Response: As explained above for the observational study, the interaction term will investigate whether associations between risk factors and outcomes differ between HIV-positive and HIV-negative groups who are included together in the same model.

- Line 23: Please describe which groups and which group comparisons will be performed.

Response: We have updated this section to read: "The primary comparison for the enteropathy substudy will be between children with HIV-SAM (group A) and SAM (group C), stratified by presence or absence of oedema. Control groups (B and D) are well-nourished children with or without HIV, to provide normative data for biomarkers and to evaluate the impact of SAM within each HIV group. Thus, biomarkers among children with HIV-SAM will first be compared to children with non-HIV SAM (to evaluate the impact of HIV) and, second, to well-nourished HIV-infected children (to evaluate the impact of SAM). Biomarkers among children with non-HIV SAM will be compared to wellnourished HIV-uninfected children."

- Line 29: Instead of Kaplan-Meier analyses and log-rank tests, I would suggest using Cox models to allow for the adjustment of possible relevant baseline differences between the comparison groups.

Response: Thank you, we will use both approaches as suggested. We have updated this sentence to read: "Time to nutritional recovery will be compared using Kaplan-Meier and log-rank tests, and Cox models to adjust for baseline differences between groups."

16) Page 47 of 49 (Supplementary Table 1) - I would suggest considering to replace the last column (time-points) with 5 columns (e.g. labelled: B, D, 12, 24, 48) and to use crosses (x) as indications for the respectively used assessments time-points to improve readability.

Response: Thank you, change made as suggested to Supplementary Tables 1 and 2 so that they are formatted in the same way.

Line 57: Please provide the location of the Blizard Institute

Response: We have amended this to "QMUL" which is defined in the footnote as Queen Mary University of London.

17) Page 49 of 49 (Supplementary table 2) - Column 'Study groups': please provide a description of the groups (A, B, C, D) below the table (instead of giving a reference to table 1).

Response: This has been added as suggested.

VERSION 2 – REVIEW

REVIEWER	André BRIEND University of Copenhagen, Denmark
REVIEW RETURNED	14-Aug-2018

GENERAL COMMENTS	This is a well written description of a study which will improve our understanding of SAM and will give important clues to improving its treatment. Minor points The group names for the enteropathy study could be chosen to make it possible to identify them without referring to the table. For instance PK for HIV positive children with oedema (kwashiorkor), PM for HIV positive children without oedema (marasmus), NK and NM for HIV negative children and NW and PW for controls or something similar. It is not clear why leg length will be measured in Zimbabwe only. As mentioned in the background, this measure is likely to be associated with long term outcome. Of note, leg length in relation to height varies with age and some age adjustment will be needed to interpret the results. See: Fredriks AM, van Buuren S, van Heel WJ, Dijkman-Neerincx RH, Verloove-Vanhorick SP, Wit JM. Nationwide age references for sitting height, leg length, and sitting height/height ratio, and their diagnostic value for disproportionate growth disorders. Arch Dis Child. 2005 Aug;90(8):807-12 Some references are not correctly justified, maybe as a result of the translation into a pdf version.
--

REVIEWER	Moses Ngari KEMRI/Wellcome Trust Research Programme, Kenya
REVIEW RETURNED	08-Aug-2018

GENERAL COMMENTS	The authors have responded to comments satisfactorily
---

REVIEWER	Stephanie Roll Institute for Social Medicine, Epidemiology, and Health Economics, Charité - Universitätsmedizin Berlin
REVIEW RETURNED	27-Aug-2018

GENERAL COMMENTS	The authors have improved the manuscript substantially. Previous concerns were adequately addressed and/or revised. I agree with the authors in that we differ in our opinions about the alpha-adjustments regarding the sample size calculation, the future data analysis and interpretation of the results in this specific exploratory study.
--

VERSION 2 – AUTHOR RESPONSE

HOPE-SAM BMJ Response to queries 16 October 2018

	Query	Response
1.	Please revise the 'Strengths and limitations' section of your manuscript (after the abstract). This section should contain five short bullet points, no longer than one sentence each, that relate specifically to the methods.	We have reduced this to 5 bullet points in total. The additional sentences under each bullet point have been removed so that each comprises a single sentence, which has been shortened from previously.
2.	The group names for the enteropathy study could be chosen to make it possible to identify them without referring to the table. For instance PK for HIV positive children with oedema (kwashiorkor), PM for HIV positive children without oedema (marasmus), NK and NM for HIV negative children and NW and PW for controls or something similar.	Thank you for this comment. We would prefer not to rename the groups in this way, because we think the notation may be confusing, and will still result in readers having to consult the Table to understand the distinction between PK and NW, for example. The group names are explained within the text, and were standardized in response to the previous round of reviews, and have also been used in the supplementary table. In addition the current WHO classification of SAM into oedematous and non-oedematous has been used in this manuscript, rather than marasmus and kwashiorkor, so overall we prefer to retain the current group names.
3.	It is not clear why leg length will be measured in Zimbabwe only. As mentioned in the background, this measure is likely to be associated with long term outcome. Of note, leg length in relation to height varies with age and some age adjustment will be needed to interpret the results. See:	We were not able to source enough knemometers for use in all 3 sites, as these have been borrowed from collaborators in Denmark. We have clarified this on page 13, when leg length is first introduced. We agree that height adjustment may be needed and thank the reviewer for the reference; we have not added details of leg

		length analysis to the current manuscript because it is outside the scope of the study overview.
4.	Some references are not correctly justified, maybe as a result of the translation into a pdf version.	We have been through the reference list and corrected these formatting errors.
5.	I agree with the authors in that we differ in our opinions about the alpha-adjustments regarding the sample size calculation, the future data analysis and interpretation of the results in this specific exploratory study.	Thank you for this comment. We have explained our rationale for alpha-adjustment in the previous revision and thank the Reviewer for accepting this difference in approach.